# Genetic mutations in GLP-1/Notch pathway reveal distinct mechanisms of Notch signaling in germline stem cell regulation

Nimmy S. John[1,2], Michelle A. Urman[1,2], Mahasin G. Mehmood[1,2,3], Vanessa Gentile[1,2,4] and ChangHwan Lee[1,2,*]

## ABSTRACT

The Notch signaling pathway is crucial for germline stem cell (GSC) regulation in *Caenorhabditis elegans*, yet the molecular and biological consequences of GLP-1/Notch mutations remain poorly understood. This study systematically analyzes commonly used and pathological *glp-1* loss- (*lf*) and gain-of-function (*gf*) mutations to investigate their effects on Notch activity at nascent transcript (ATS), mRNA, and germline levels. Using complementary direct readouts of Notch activation, including *sygl-1* activation sites, mRNA levels, and germline functional assays of the Notch-responsive GSC pool and progenitor zone (PZ), we demonstrate that the severity of *glp-1* mutations is dependent on their position within the GLP-1 protein. Among the commonly used *glp-1* alleles we examined, NICD mutations reduced Notch transcriptional activation at cellular and germline levels while having little impact at the chromosomal (ATS) level, whereas partial *lf* NECD mutations have minimal effects across all biological levels. Furthermore, a series of regression analyses of *sygl-1* activation, mRNA production, and PZ size reveal strong correlations, qualifying these readouts as predictive markers for germline function. These findings provide a comprehensive framework for understanding *glp-1* mutation effects and offer new insights into the regulation of Notch signaling in stem cell biology.

KEY WORDS: *Caenorhabditis elegans* gonad, Notch signaling, *Glp-1* Notch receptor, Transcriptional regulation, Spatial pattern analysis, *Sygl-1*, Gradient, Notch mutations

## INTRODUCTION

Timely regulation of Notch signaling, a major regulator of animal development and homeostasis, is crucial for proper cell fate decisions, cellular functions, and tissue patterning (Albert Hubbard and Schedl, 2019; Artavanis-Tsakonas et al., 1999; Crittenden et al., 2019; Greenwald et al., 1983; Wilkinson et al., 1994). Notch signaling is transduced by the physical interaction between the Notch ligands and the Notch receptors, which in turn activates the transcription of target genes (Kopan and Ilagan, 2009). Notch receptors, a key core component of the signaling pathway, have been

[1]Department of Biological Sciences, University at Albany, State University of New York, Albany, NY 12222, USA. [2]The RNA Institute, University at Albany, State University of New York, Albany, NY 12222, USA. [3]School of Podiatric Medicine, Temple University, Philadelphia, PA 19107, USA. [4]Department of Biology, The State University of New York at New Paltz, New Paltz, NY 12561, USA.

*Author for correspondence (chlee@albany.edu)

N.S.J., 0000-0001-5094-2918; M.A.U., 0000-0001-8812-9279; C.L., 0000-0002-9821-9312

linked to various human diseases, including dilated cardiomyopathy, T-ALL, Alagille Syndrome, CADASIL, and various types of cancer (Bolós et al., 2007; Dang et al., 2000; Donahue and Kosik, 2004; Gridley, 2003; McDaniell et al., 2006; Mutvei et al., 2015; Penton et al., 2012; Wang et al., 2020; Weng et al., 2004; Yamamoto, 2020; Zhou et al., 2022). Despite extensive studies on Notch, the disease mechanisms of many of these mutations are still not determined or even whether they cause gain- or loss-of-function (*gf* or *lf*) Notch activity. Here, we use the *Caenorhabditis elegans* germline to reveal the pathological mechanisms and molecular consequences of the commonly used Notch mutations that are relevant to human diseases.

*C. elegans* expresses two Notch receptors, GLP-1 and LIN-12. Notably, only GLP-1 is expressed in the germline while only LIN-12 is in the vulva, although both share the mechanistic pathway involving direct ligand-receptor interaction and subsequent transcriptional activation of Notch target genes (Kopan and Ilagan, 2009). Specifically in the distal germline, the transmembrane Notch receptor GLP-1 is cleaved upon its interaction with Notch ligands on the distal tip cell (DTC), such as LAG-2 and APX-1, which induces proteolytic cleavage of GLP-1 to release the Notch intracellular domain (NICD). The NICD then translocates to the nucleus to activate transcription of two Notch targets, *sygl-1* and *lst-1*, stem cell effectors for germline stem cell (GSC) maintenance (Fig. 1A,B) (Austin and Kimble, 1987; Chen et al., 2020; Lee et al., 2016).

GLP-1 contains several broadly conserved domains, including the epidermal growth factor (EGF) repeats, which are exposed outside the cell as a part of the Notch extracellular domain (NECD), and the ankyrin repeats (ANK) in the NICD (Arruga et al., 2018; Kopan and Ilagan, 2009) (Fig. 1B and Fig. S1A). Similar to disease-causing Notch mutations in humans, *glp-1* mutants often lead to pathologies, such as GSC loss, germline tumors, infertility, or reproductive decline, varying in severity and onset timing (Austin and Kimble, 1987; Berry et al., 1997; Crittenden et al., 2019; Kodoyianni et al., 1992; Pepper et al., 2003). Despite numerous studies, *glp-1* mutations are often characterized by broad phenotypic assessments, such as brood size or pharyngeal induction, and crudely categorized as *lf*. This limits our understanding of the molecular roles of individual GLP-1 domains in their molecular and cellular functions, including transcriptional activation and GSC regulation (Berry et al., 1997; Kodoyianni et al., 1992; Pepper et al., 2003). Recent studies using advanced, precise assays, such as single-molecule RNA visualization or high-resolution protein localization, suggest that each *lf* or *gf glp-1* mutation affects distinct aspects of Notch-induced transcription, resulting in distinctive pathological phenotypes (Lee et al., 2016).

Under normal conditions, Notch-induced transcriptional activation is spatially regulated within the GSC pool in a graded manner. The distal end of the germline, adjacent to the Notch-signaling DTC, exhibits the highest *sygl-1* and *lst-1* transcriptional activity (Crittenden et al., 2019; Lee et al., 2016). This gradient is essential for establishing germline polarity and maintaining the GSC pool, a mechanism also observed broadly across species and tissues, including *Drosophila*

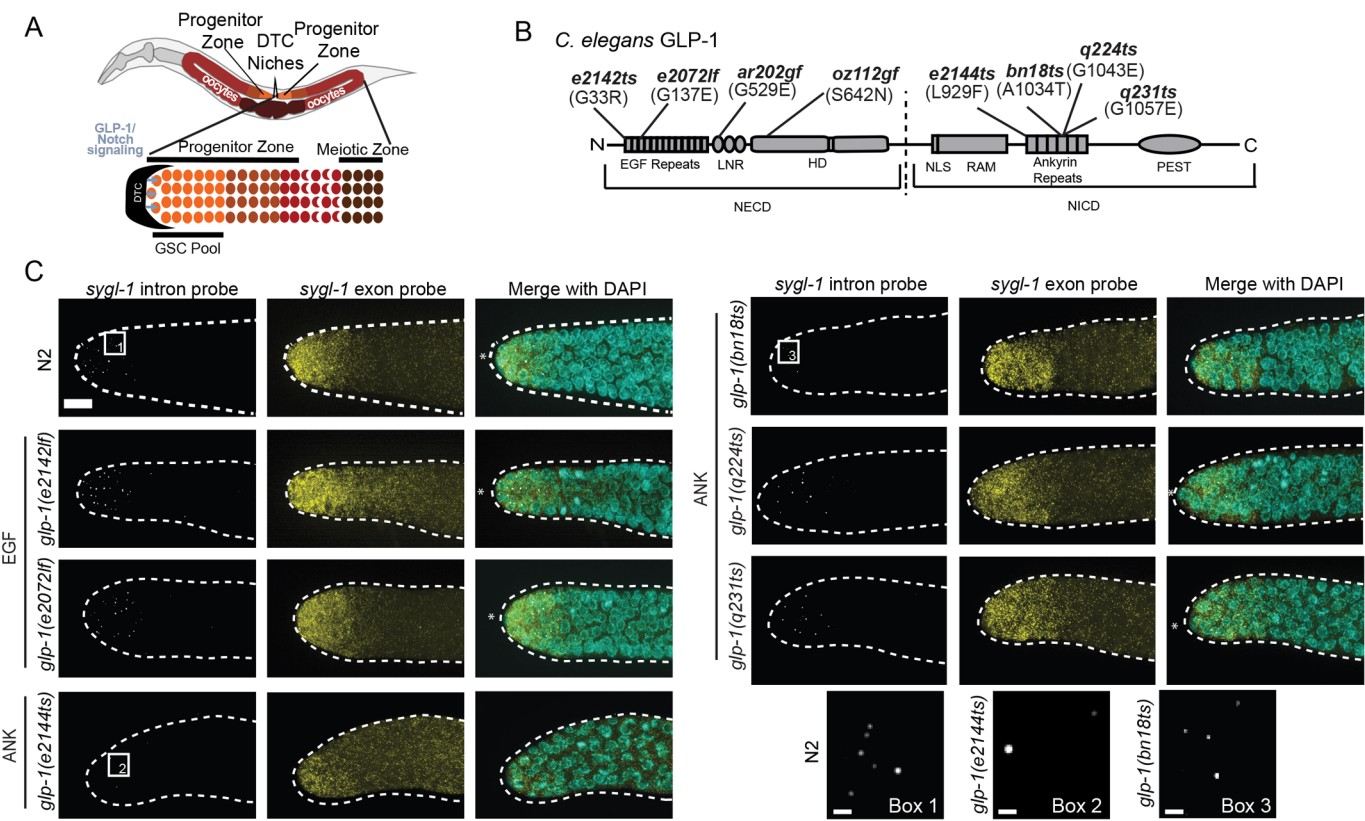

**Fig. 1. Visualizing Notch-induced transcriptional response in *glp-1* loss-of-function mutants.** (A) A schematic of adult *C. elegans* hermaphrodite with a focus on two U-shaped gonads. The distal tip cell (DTC, germline stem cell niche) is located at each distal end (black crescents). Germline stem cells (GSCs) are indicated in orange. Differentiating cells are in brown, and oocytes are in dark brown. (B) Schematic of GLP-1 structure, with its domains indicated. EGF, epidermal growth factor-like repeats; LNR, *lin-12*-Notch repeats; HD, heterodimerization domain; NLS, nuclear localizing signal; RAM, RBPJκ Associated Module; ANK, ankyrin repeats; and PEST, PEST sequence. NECD indicates the Notch extracellular domain, and NICD indicates the Notch intracellular Domain, separated by a dashed line. The glp-1 mutations relevant to this study are indicated with their allelic names. (C) Z-projected *sygl-1* smFISH images of the distal gonad of *glp-1* mutants. Two smFISH probe sets visualize nuclear *sygl-1* nascent transcripts at the active transcription sites (ATS; white) and cytoplasmic mRNAs (yellow). DAPI marks DNA (cyan). Scale bar: 10 μm for gonadal images and 1 μm for magnified images (Boxes 1-3).

germarium and the human left ventricle (Dansereau and Lasko, 2008; Lee et al., 2016; Rochais et al., 2009). However, how GLP-1/ Notch signaling generates and regulates this gradient under different physiological or pathological conditions remains unclear. Is it determined by the amount of activated GLP-1 (and consequently NICD abundance), the strength of NICD in forming the Notch transcription factor complex (NTFC) and its stability, or other factors or modifications influencing Notch-induced transcription activity?

Here, we perform a comprehensive comparative analysis of eight commonly used, pathological *glp-1* mutations, each containing a genetic lesion in one of four distinct GLP-1 domains (Table 1 and Fig. 1B) (Berry et al., 1997; Kodoyianni et al., 1992; Pepper et al., 2003). We re-classify these mutations based on their effects on the spatial regulation of Notch-induced transcription, Notch-responsive GSC pool size and activity, and germline function. Through correlation analyses and regression modeling, we reveal links between Notch signaling strength, transcriptional activation, GSC activity, and germline function such as fertility, enabling the prediction of genetic and phenotypic traits of untested mutations.

## RESULTS

### Notch activity is systematically analyzed in each *glp-1* mutant using direct Notch transcriptional readouts

To systematically examine *glp-1* mutations, we selected eight commonly used, mutations that are relevant to human diseases,

including six partial-*lf* and two *gf* mutations, all of which distinctively affect embryogenesis, germline function, progenitor zone lengths, and egg-laying rates (Table 1) (Berry et al., 1997; Fox and Schedl, 2015; Kodoyianni et al., 1992; Pepper et al., 2003). To investigate the mechanisms underlying pathological *glp-1* mutations and their biological consequences, we used single-molecule RNA fluorescence *in situ* hybridization (smFISH) targeting *sygl-1*, a direct Notch target that has been used as a precise readout of Notch activation and a GSC indicator (Crittenden et al., 2019; Lee et al., 2016; Lynch et al., 2022; Parker et al., 2021; Raj and Tyagi, 2010; Raj and van Oudenaarden, 2009; Urman et al., 2024). In this study, *sygl-1* smFISH was performed with two probe sets, each targeting either the introns or exons of *sygl-1*, allowing us to visualize both nascent transcripts (pre-mRNAs) and cytoplasmic mRNAs at the single RNA molecule level, as previously confirmed (Fig. 1C, Figs S1B, and S2A) (Lee et al., 2016, 2017; Raj and van Oudenaarden, 2009). As expected, we observed several bright spots in the distal gonad with the intron-specific probe set, indicating the *sygl-1* active transcription sites (ATS), where many nascent transcripts are actively produced (Fig. 1C, '*sygl-1* intron probe'). We also confirmed that the exon-specific probe set marks individual mRNAs (dim, small yellow spots) as well as ATS (bright yellow spots), which overlapped with the intron probe signals, all as previously reported (Lee et al., 2016). Because approximately half of the mutations analyzed are either temperature-sensitive mutants

**Table 1. glp-1 mutations used in this study**

| Mutation | GLP-1 domain | Amino acid change | Temperatures (permissive, restrictive) | Reference |
|---|---|---|---|---|
| glp-1(q224ts) | Ankyrin repeat | G1043E | 15°C, 25°C | Kodoyianni et al. 1992 |
| glp-1(q231ts) | Ankyrin repeat | G1057E | 15°C, 25°C | Kodoyianni et al. 1992 |
| glp-1(bn18ts) | Ankyrin repeat | A1034T | 15°C, 25°C | Kodoyianni et al. 1992 |
| glp-1(e2144ts) | Ankyrin repeat | L929F | 15°C, 25°C | Kodoyianni et al. 1992 |
| glp-1(oz112gf) | HD domain | S642N | 20°C | Berry et al. 1997 |
| glp-1(ar202gf) | LNR domain | G529E | 15°C, 25°C | Pepper et al. 2003 |
| glp-1(e2072lf) | EGF repeats | G137E | 20°C | Kodoyianni et al. 1992 |
| glp-1(e2142ts) | EGF repeats | G33R | 15°C, 25°C | Kodoyianni et al. 1992 |

(*ts*) or require growth temperatures different from the standard 20°C, we performed *sygl-1* smFISH across temperatures ranging from 15°C to 22.5°C (Fig. S1B-D). 22.5°C was used for *ts* mutants that completely lose the germline at their restrictive temperature, 25°C. The total number of *sygl-1* ATS in the distal gonad, reflecting the overall Notch-induced transcriptional activation, and their spatial distribution, reflecting the gradient pattern of Notch activation, remain largely the same across temperatures, indicating that Notch activation is not significantly affected by temperature variation (Fig. S1B-D). Therefore, for all subsequent analyses, we used N2 worms maintained at 20°C as a control for *glp-1* mutants.

### Each glp-1 lf mutant uniquely affects Notch-induced transcription yet can be broadly classified into three categories

We first analyzed *lf* mutations, which are located in two highly conserved domains: the ANK within the NICD and EGF repeats (EGF) within the NECD. Mutations in the ANK domain likely affect the activity or stability of the NICD, while the mutations in the EGF domain may alter the amount of NICD cleaved and translocated into the nucleus (Kopan and Ilagan, 2009). We excluded *null* mutants from this study as they quickly deplete the germline, leaving no GSCs to examine at the young adult stage (YA or day 1) (Austin and Kimble, 1987; Kodoyianni et al., 1992). First, to analyze the tissue-level effects of *glp-1* mutations, we compared the total number of Notch-responsive GSCs containing at least one *sygl-1* ATS, which estimates the GSC count, across the six *lf* mutants both at their permissive and restrictive temperatures (Fig. 2A, Fig. S2B). Here, we focus on the effects of these mutations at their restrictive or semi-restrictive temperatures, where most mutants exhibit severe defects in Notch signaling response (e.g. Fig. 2A versus Fig. 2B). For *glp-1* mutants that become sterile at the fully restrictive temperature [e.g. *glp-1(e2142)* at 25°C] and therefore completely lose GSCs and *sygl-1* ATS, we instead analyzed the mutants at semi-restrictive temperature of 20 or 22.5°C. As expected, most mutants exhibited a reduced number of ATS-containing cells, confirming that *glp-1* mutations diminish Notch signaling activity and GSC maintenance. The mutations located in the ANK drastically reduced the Notch-responsive GSC number regardless of the location or type of the mutation, though to varying degrees. In contrast, mutations in the EGF had little to no effect on the Notch-responsive GSC number, although they have been classified by *lf* mutants (Kodoyianni et al., 1992) (Fig. 2A). The trend remained essentially the same for the ATS extent, which measures the GSC pool size, and its ratio to the total number of *sygl-1* ATS in the distal gonad, which estimates the density of ATS-containing cells within the GSC pool (Fig. 2B,C, Figs S2C,D, S3A).

We then determined how each *glp-1* mutation affects Notch activation at the cellular level by examining the number of *sygl-1*

ATS in each Notch-responsive GSC (Fig. 2D,E, Fig. S2D). Consistent to the tissue-level changes (Fig. 2A-C), the ratio of GSC containing more than one *sygl-1* ATS and average number of ATS in a Notch-responsive GSC greatly decreased with mutations in the ANK, whereas they remain essentially unchanged with mutations in the EGF, suggesting that the ANK – and potentially the NICD – may have a stronger effects on Notch activity regulation than the NECD under partially restrictive conditions (Fig. 2D,E, Fig. S3B). Also, the progenitor zone (PZ) size, which estimates the number of mitotic germ cells by measuring the distance between the distal end and the cells with a crescent-shaped meiotic nucleus and therefore reflects the potential for generating new gametes, exhibited a similar trend (Fig. S3C). This trend persisted in *glp-1* mutants at the permissive temperature, albeit to a lesser extent (Figs S2F and S3C). Similarly, the total number of *let-858* ATS, a Notch-independent gene constitutively expressed in the germline (Lee et al., 2016), also decreased at the distal gonad by most of the *glp-1* mutations, suggesting that these mutations have direct or indirect effects on overall transcriptional level in the GSCs (Fig. S4A,B).

To analyze the impact of each *glp-1* mutation on Notch activity at the chromosomal level, we measured the signal intensities of individual *sygl-1* ATS, which reflects the number of nascent transcripts produced at each *sygl-1* locus (Fig. 2F and Fig. S2G). Although the ANK mutations led to a greater reduction in ATS intensity than EGF mutations, the decline was considerably smaller compared to changes observed at the tissue and cellular levels (Fig. 2A,F,G, Fig. S2G). The summed ATS intensity, representing overall Notch-induced *sygl-1* transcriptional activity within a nucleus, followed a similar trend, albeit with a slightly greater reduction in ANK mutants due to a lower average ATS number per cell (Fig. 2F,G, Fig. S2G,H). Nuclear size is a reliable indicator of germ cell cycle stage, as it gradually increases with cell cycle progression until M phase (Lee et al., 2016; Seidel and Kimble, 2015). The average nuclear size in the distal gonad remained unchanged across wild-type and *glp-1* mutants, indicating that these mutations are unlikely to activate cell cycle checkpoints (Fig. 2H). To further validate our findings, we analyzed a second Notch target, *lst-1*, in EGF and ANK mutants (Fig. S13). *lst-1* transcription showed a very similar trend to *sygl-1*, though with overall lower ATS numbers and intensities as expected (Lee et al., 2016), thereby reinforcing our conclusions with *sygl-1* (Fig. S13A-C).

### sygl-1 mRNAs reflect the changes in ATS observed in glp-1 mutants

Along with *sygl-1* ATS marking Notch-responsive GSCs, its cytoplasmic mRNAs serve as an additional Notch readout, indicating the size and distribution of the GSC pool. *sygl-1* mRNAs exhibit a spatially graded pattern similar to ATS in the

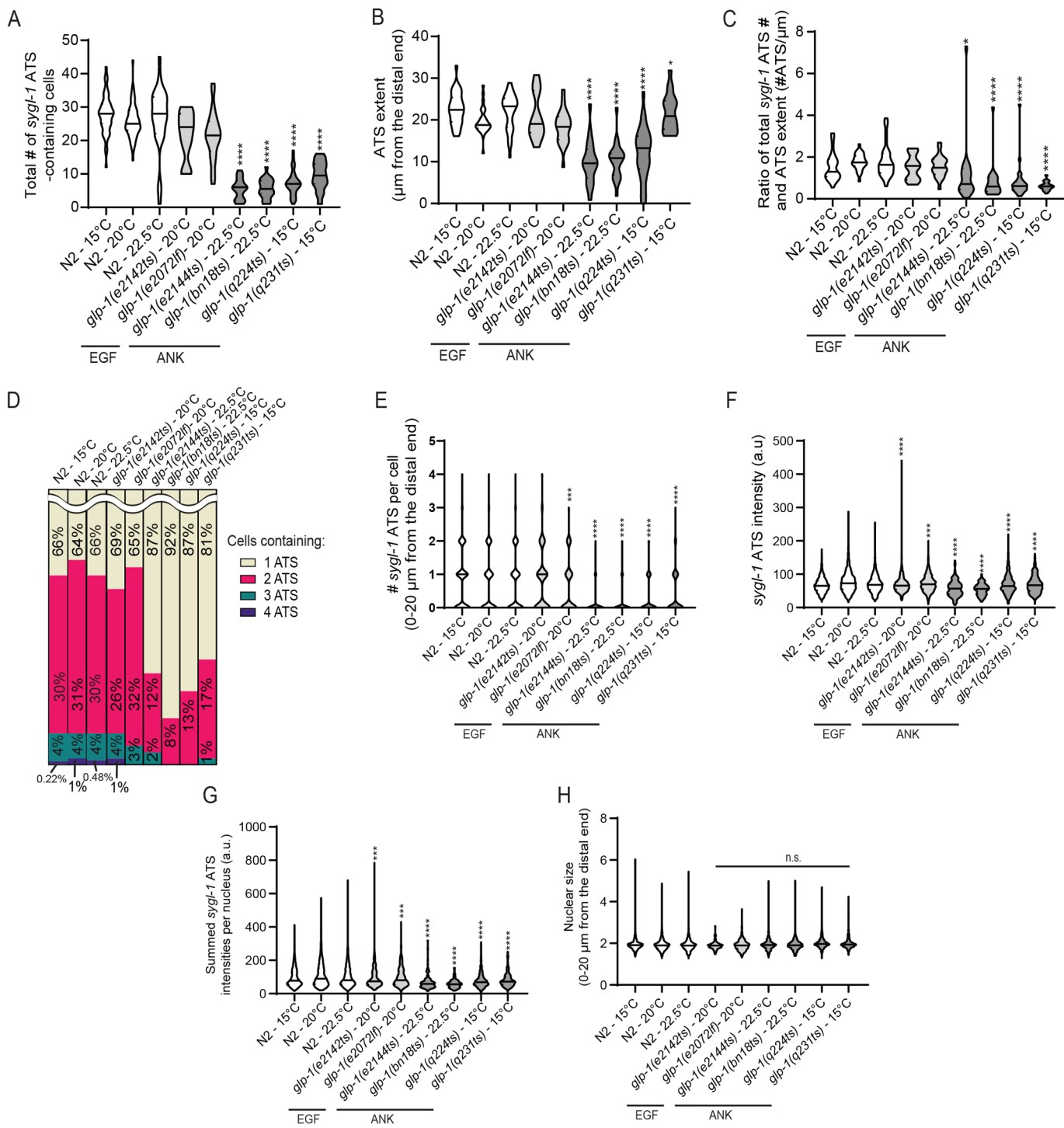

**Fig. 2. Analyzing *sygl-1* transcriptional activation in *glp-1* mutants.** (A) Number ATS containing cells, (B) *sygl-1* ATS extents (distance between farthest two ATS), and (C) the ratio of the total ATS numbers and the ATS extent in the corresponding gonad are measured in glp-1 mutants at their restrictive temperatures. (B,C) *n*=32, 13, 23, 23, 24, 30, and 20 gonads in the order displayed. (D) The percentages of germ cells containing 1, 2, 3, or 4 *sygl-1* ATS in each mutant. (E) The number of *sygl-1* ATS per cell (ranges 0-4) in each mutant. (A,D,E) *n*=62, 13, 24, 23, 24, 67, and 30 gonads in the order displayed. (F) Individual *sygl-1* ATS intensities in each mutant. *n*=2398, 391, 709, 152, 148, 546 and 336 ATS. (G) Summed *sygl-1* ATS intensities in each mutant. *n*=1592, 278, 513, 126,136, 484, and 281 nuclei. (H) The nuclear size (radius) in each mutant. *n*=2970, 420, 1062, 801, 1038, 1841 and 776 nuclei. (A-C, E-H) Throughout this study, the *P*-values for ANOVA are indicated in asterisks. \**P*<0.05; \*\**P*<0.01; \*\*\**P*<0.001; \*\*\*\**P*<0.0001; and 'n.s.': non-significant. Each *glp-1* mutant data are compared to N2 (wild type) at the corresponding temperature.

gonad at Day 1, although mRNAs extend a few germ cell diameters (gcd) further into the proximal gonad due to their longer half-life compared to ATS (~1 h versus ~10 min) (Lee et al., 2016; Lynch et al., 2022; Urman et al., 2024). To analyze the tissue-level effects of *glp-1* mutations on *sygl-1* mRNAs, we quantified the total number of *sygl-1* mRNAs in the distal gonad of each mutant. The

observed trend closely mirrored that of ATS, with ANK mutations exhibiting a significant reduction in both ATS and mRNAs (Figs 2A, 3A, Fig. S5A-D). This trend persisted when analyzing the number of *sygl-1* mRNAs per cell in the distal gonad (Fig. 3B, Fig. S5E,F), which reflects Notch-induced transcription at the cellular level. Additionally, the ratios of ATS to mRNAs in the distal gonad, which estimates the number of *sygl-1* mRNAs produced per ATS, followed a similar trend (Fig. 3B,C). The *sygl-1* mRNA extent, which complements the ATS extent for estimating the GSC pool size, showed essentially the same trend to that of ATS extent with *glp-1* mutants (Figs 2B, 3D). The ratio of ATS extent to mRNA extent was similar across wild-type and *glp-1* mutants, indicating a tightly regulated relationship between Notch-induced *sygl-1* ATS and mRNAs (Fig. 3E). Our analysis with *lst-1* showed similar results, confirming our conclusions with *sygl-1* (Fig. S12F,G).

### *glp-1* mutations alter the spatial pattern of *sygl-1* transcriptional activation

Notch-induced *sygl-1* transcription is tightly regulated within the GSC pool to create a steep gradient in probability of Notch activation, which is crucial for establishing germline polarity and defining the GSC pool boundary (Crittenden et al., 2019; Lee et al., 2016; Lynch et al., 2022; Urman et al., 2024). Given that each *glp-1* mutation reduces Notch activities at both the tissue and cellular levels to varying extents, we anticipated that these mutations would

distinctively affect the *sygl-1* transcriptional pattern in the distal gonad. To test this idea, we measured the percentage of GSCs containing at least one *sygl-1* ATS as a function of distance from the distal end of the gonad along its lateral axis and compared these values across the *glp-1 lf* mutants (Fig. 4A). This data estimates the probability of Notch-induced transcriptional activation at varying distance from the distal end of the gonad, where the DTC resides. As previously reported, the wild type exhibited a steep gradient of Notch activation probability within the first 25 µm, corresponding to the Notch-responsive GSC pool region (Lee et al., 2016; Lynch et al., 2022; Urman et al., 2024) (Fig. 4A, 'N2'). Each *glp-1* mutant reduced this gradient to varying degrees, consistent with the tissue- and cellular-level changes in *sygl-1* ATS and mRNA numbers, altering the Notch-responsive GSC pool size (Figs 2A, 3A, 4A, red-dashed lines). The EGF mutants did not alter the Notch activation pattern, similar to their unchanged ATS and mRNA numbers (Figs 2A, 3A, 4A), whereas the ANK mutants significantly reduced it, shrinking the GSC pool size by up to 40% (Fig. 4A, red-dashed lines). Notably, the gradient pattern persisted in most *glp-1* mutants, except for *glp-1(q231)*, which exhibited a plateau in the probability of Notch activation within the GSC pool (Fig. 4A and Fig. S6A). This trend remained for the number of *sygl-1* ATS per Notch-responsive GSC plotted against the distance from the distal end, which reflects the spatial pattern of the number of *sygl-1* loci actively responding to Notch signaling, an indicator of cellular-level

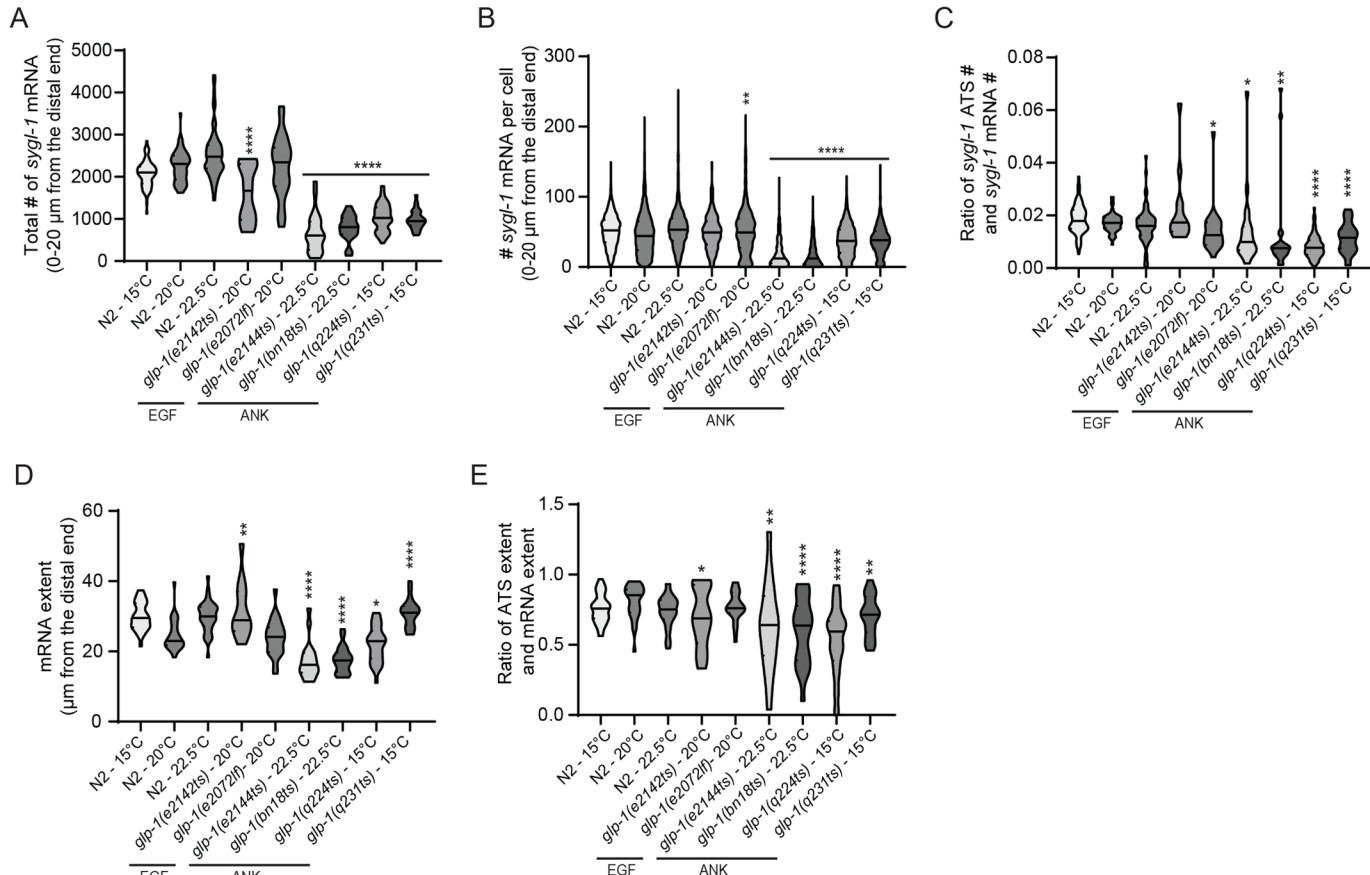

**Fig. 3. Assessing how *glp-1* mutations influence Notch-induced transcriptional activity using *sygl-1* mRNA readouts.** (A,B) The total number of *sygl-1* mRNA (0-20 µm from the distal end) and the total number of *sygl-1* mRNA per cell (0-20 µm from the distal end) are measured in each *glp-1* mutant. *n*=2970, 420, 1062,801, 1038, 1841 and 776 nuclei in the order displayed. (C) The ratio of the total number of *sygl-1* ATS and the total number of mRNA. *n*=32, 13, 23, 23, 24, 30, and 20 gonads. (D) *sygl-1* mRNA extent measured from the distal end. (E) The ratio of the ATS extent and mRNA extent in the corresponding gonads. (D,E) *n*=32, 13, 23, 23, 24, 30 and 20 gonads.

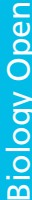

**Fig. 4. Analyzing the spatial pattern of Notch-induced *sygl-1* transcription in *glp-1* mutants.** (A-C) The percentages of cells containing *sygl-1* ATS (A), the number of *sygl-1* ATS per cell (B), and the number of *sygl-1* mRNA per cell (C) are plotted as a function of the position from the distal end to reveal their spatial patterns in the germline. The red dashed line indicates the GSC pool. *n*=20, 62, 20, 13, 24, 23, 24, 67, and 30 gonads in the order displayed. Error bar: the standard error of the mean (s.e.m.).

Notch activation (Fig. 4B and Fig. S6B). Another Notch readout, *sygl-1* mRNAs, exhibited the same spatial patterns across all corresponding *glp-1* mutants, whether assessed by their numbers per cell or the percentage of cells containing *sygl-1* mRNAs (Fig. 4C, Fig. S6C-E), confirming the trend observed in the spatial analyses of ATS (Fig. 4A,B). Notably, the plateaued Notch activation pattern observed with ATS in *glp-1(q231)* persisted in the mRNA spatial analyses, further supporting the role of the mutated region in regulating the Notch transcriptional activation gradient [Fig. 4A-C, *glp-1(q231)*]. This trend was also observed in the spatial pattern of intensities at the chromosomal and cellular levels in all *glp-1 lf* mutants as well as the spatial patterns of another Notch target, *lst-1* (Fig. S7A,B and S12H-J). These results indicate a linear relationship between Notch-induced *sygl-1* ATS and mRNAs, where changes in ATS directly translate to proportional changes in mRNA levels, allowing predictions of one based on the other. This relationship will be further discussed in the discussion.

### *glp-1 gf* mutations increase the overall Notch transcription in the germline, but not at the ATS and mRNA levels

To further understand how *glp-1* mutations affect Notch transcriptional activation, we now turn to two commonly used *glp-1 gf* mutations located in the NECD, which mimic human Notch mutations causing several types of cancers, including reproductive and lung cancer (Dang et al., 2000; Mutvei et al., 2015; Wang et al., 2020) (Fig. 5A). Given that both *gf* mutations are located in the NECD, they are likely to affect the abundance of the NICD cleaved and translocated into the nucleus (Fig. 5A). As expected, *sygl-1* smFISH revealed a significant increase in overall tissue-level *sygl-1* transcription compared to wild type, with *sygl-1* ATS and mRNA levels being at least four times higher in the distal gonad (Fig. 5C-I, Fig. S5F,G). We confirmed that *glp-1(ar202)* is proliferative mutant, where Notch transcriptional activation extends to a more proximal region, expanding the Notch-responsive GSC pool (Fig. 5B-D). In contrast, *glp-1(oz112)* induces a tumorous germline, with Notch transcriptional activation extending beyond 100 µm from the distal end of the gonad, eliminating the pachytene region (Fig. 5B-D). This was consistent when analyzing the PZ size in these *glp-1 gf* mutations (Fig. S6H). However, when focusing on the first 20 µm from the distal end of the gonad, where the majority of Notch-responsive *sygl-1* transcription occurs in the wild-type gonad, the number of *sygl-1* ATS-containing GSCs, ATS per cell, individual ATS intensities, and other features reflecting cellular- and chromosomal-level Notch activity remained unchanged or slightly decreased in both *glp-1 gf* mutants (Fig. 5E-H, Fig. S6I). Consistently, *sygl-1* mRNA density and the number of mRNA per GSC also decreased at the distal gonad (Fig. 5H,I). However, the average nuclear size of GSCs was unaffected by the *gf* mutations, suggesting no activation of cell cycle checkpoints (Fig. 5J). The *sygl-1* spatial pattern analyses revealed that both *sygl-1* ATS and mRNAs are extended more proximally, increasing the overall probability of Notch transcriptional activation, similar to the tissue-level assays (Fig. 5K-M, and Fig. S6J). However, the probability remained unchanged or slightly decreased at the distal gonad region (0-20 µm from the distal end), consistent with the cellular-level assessment (Fig. 5K-M, and Fig. S6J). This trend was also observed in the spatial pattern of intensities at the chromosomal and cellular levels in all *glp-1 gf* mutants (Fig. S7A,B).

### *sel-12*/Presenilin mutants phenocopy the *glp-1* EGF mutants

Our results with *glp-1* partial *lf* and *gf* mutants demonstrate that mutations in the NICD, but not the NECD, can significantly alter the

Notch transcriptional activation pattern and its activity under partially restrictive conditions (Figs 2–5). These suggest that changes in the abundance of NICD are less effective than changes in its activity or stability. To confirm this, we examined Notch-induced *sygl-1* transcription within two *sel-12* mutations, which mimic the human Presenilin mutations causing dilated cardiomyopathy (Eimer; Okochi et al., 2000) (Fig. 6A). SEL-12 is the catalytic subunit of γ-secretase and is orthologous to Presenilin in humans (Fig. S8A) (Okochi et al., 2000). *sel-12 lf* mutations typically result in a defect in endoproteolysis, decreasing the amount of NICD produced (Okochi et al., 2000). Similar to the mutations in the NECD (EGF *lf* mutations and *glp-1 gf* mutations), both *sel-12* mutations minimally altered *sygl-1* ATS and cytoplasmic mRNA in terms of their numbers, intensities, and spatial patterns, supporting that NICD abundance is a less significant contributing factor for regulating Notch-induced transcriptional activation than NICD stability or activity (Fig. 6B-G, Fig. S8B-G). Given the partial redundancy between *sel-12* and its paralog *hop-1* (Li and Greenwald, 1997), additional studies targeting either or both genes are needed to validate these findings.

### Comparative analyses reveal the links between the location and type of *glp-1* mutations, Notch transcriptional activation, and germline function

We observed remarkably similar trends in *sygl-1* ATS and mRNA patterns across all *glp-1* and *sel-12* mutants. This consistency suggests a simple, linear relationship between ATS and mRNA levels, which may extend to the biological consequences of transcription altered by these mutations, including changes in PZ size, an indication of germline activity. To confirm this, we conducted a series of comparative regression analyses using various measurements of Notch transcriptional readouts obtained in this study (Figs 2–7). First, we built a regression model to describe the relationship between the total number of *sygl-1* ATS in the gonad, which reflects the overall tissue-level Notch transcriptional activation, and the number of mRNAs per GSC, an estimate of cellular-level Notch activity, across *glp-1* mutants (Fig. 7A,B, and Fig. S9A). Both the combined dataset of all *glp-1* mutants and the individual datasets for each mutant were best fitted to a simple linear regression model, which indicates that there is a strong correlation between ATS number and cellular mRNA number and that the tissue-level Notch activation can reliably predict cellular-level Notch activity (Fig. 7A,B, Figs S9A and S10). Notably, the slopes of the regression models differed between EGF and ANK mutants; however, individual mutants within each group exhibited similar slopes (Fig. 7A,B, Figs S9A and S10). This trend aligns with all other ATS and mRNA analyses (Figs 2–5), further confirming the idea that EGF and ANK mutations differentially influence Notch transcriptional activation and its spatial pattern. Complementary to these results, we constructed a regression model to examine the relationship between the percentage of *sygl-1* ATS-containing GSCs, which reflects the probability of Notch activation, and the percentage of *sygl-1* mRNA-containing cells, estimating the cellular-level Notch activity, for each *glp-1* mutation (Fig. 7C, Figs S9B and S11). The datasets from the wild type and all *glp-1* mutants fit to a hyperbolic curve with a strong correlation, indicating that one can effectively predict the other (Fig. 7C, Figs S9B and S11). This tight correlation also extended to the relationship between overall cellular-level *sygl-1* transcriptional activity, reflecting the total nascent *sygl-1* transcripts produced per GSC, and the number of *sygl-1* cytoplasmic mRNAs per GSC, revealing another link between the Notch transcriptional readouts (Fig. 7D, Figs S9C and S12).

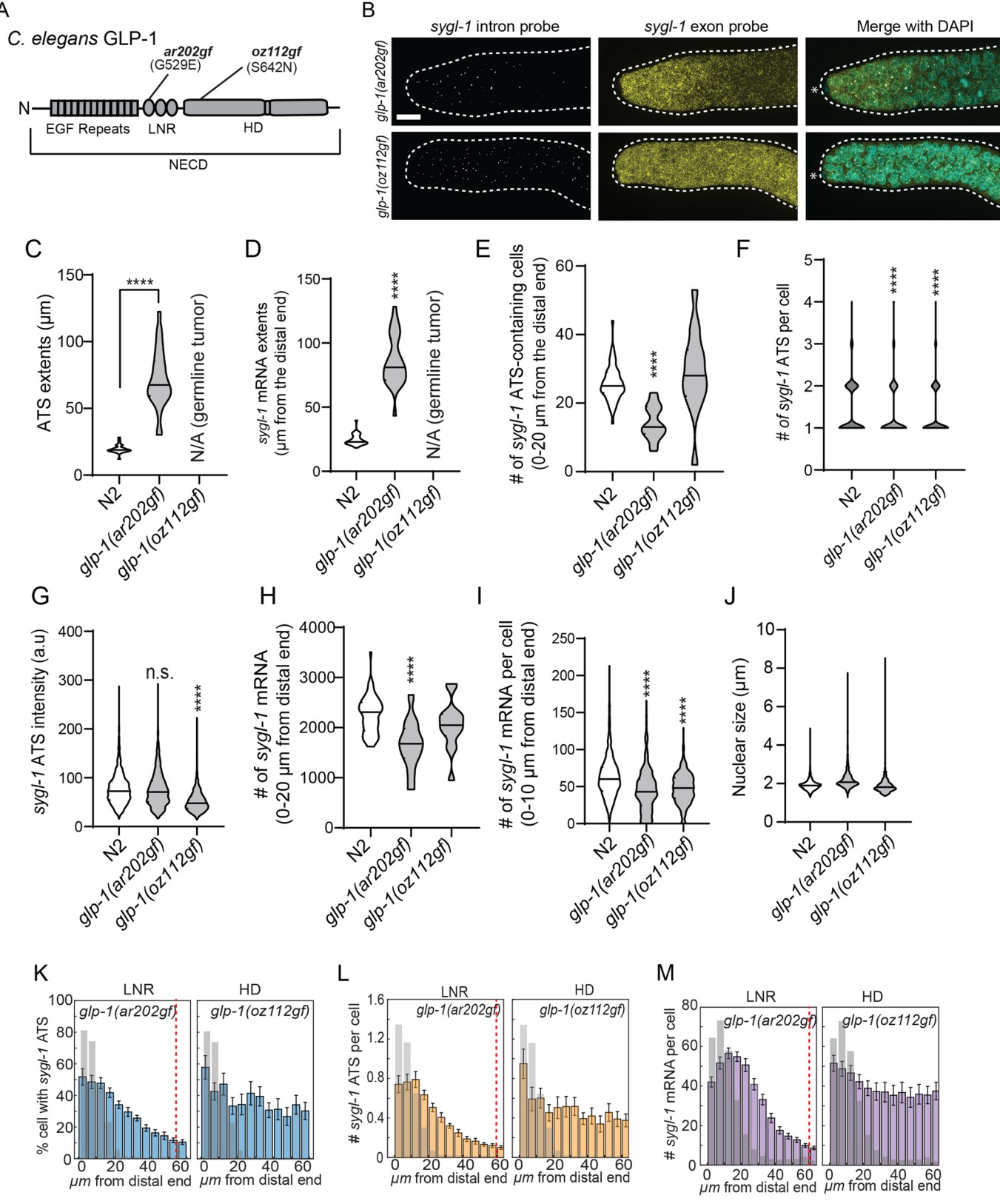

**Fig. 5. Analyzing *glp-1 gf* mutants.** (A) Two *glp-1 gain-of-function (gf)* mutations analyzed at 20°C in this study are indicated. (B) Z-projected *sygl-1* smFISH images of *glp-1 gf* mutants. (C-J) *sygl-1* transcriptional activation and its ATS, mRNA, and germline-level activity are assessed as described in Figs 2 and 4. (C,D) *n*=32, 28, and 14 gonads in the order displayed. (E,F, J-M) *n*=62, 29, and 14 gonads. (G) *n*=2398, 1808, and 2882 ATS. (H,I) *n*=2970, 957, and 643 nuclei. (K-M) Red dashed line: the GSC pool boundary.

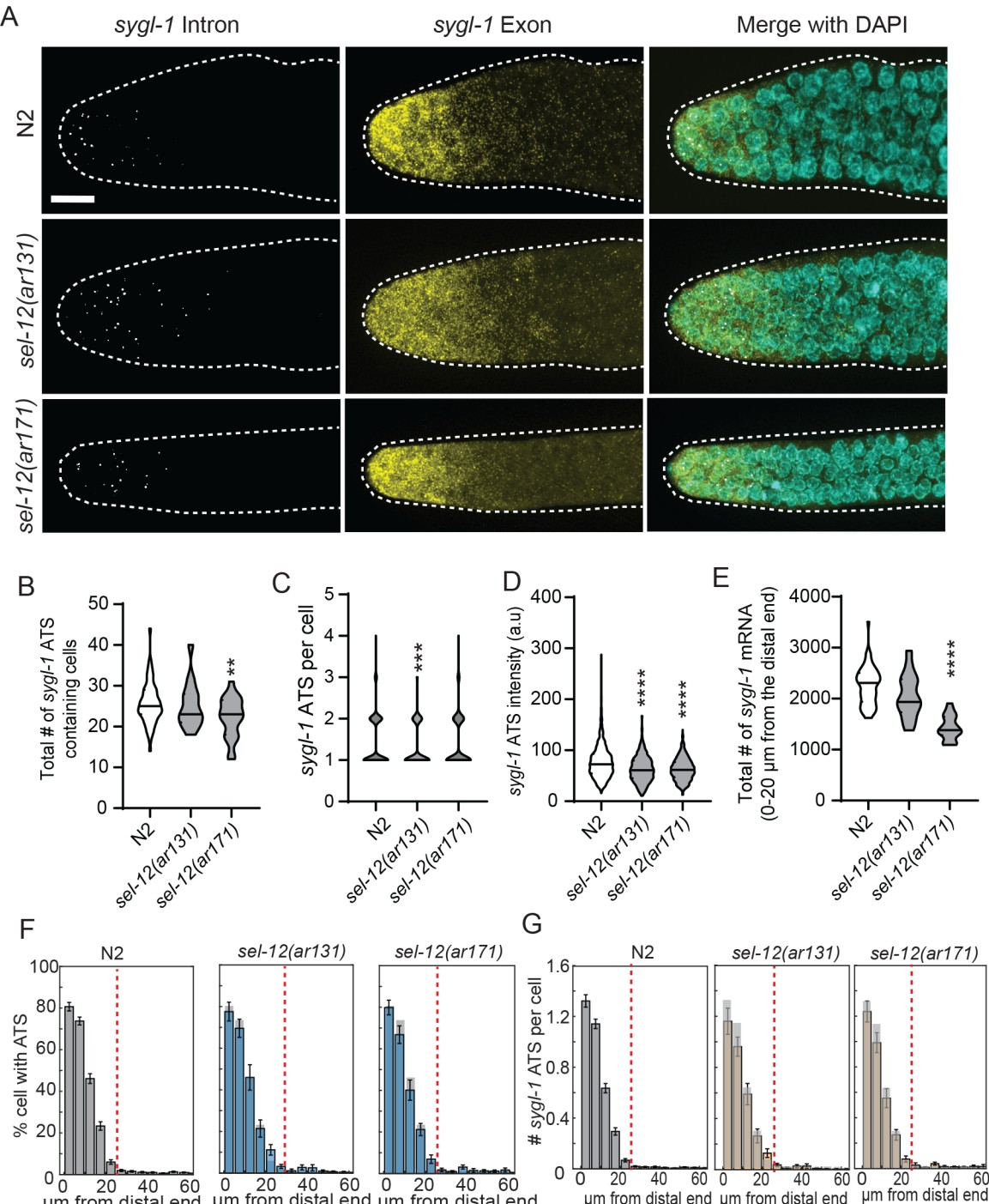

**Fig. 6. Analyzing *sygl-1* transcriptional response in *sel-12* mutants.** (A) Z-projected *sygl-1* smFISH images of *sel-12* mutants grown at 20°C. (B-G) *sygl-1* transcriptional activation and its ATS and mRNA spatial patterns in *sel-12* mutants. *n*=2397, 504, and 698 ATS from at least 20 gonads per strain. (F,G) Red dashed line: the GSC pool boundary. Scale bar: 5 μm. Error bar: SEM.

Further regression analyses between these Notch readout values and PZ size, an indicator of germline activity and the capacity to generate new gametes, revealed hyperbolic relationships (Fig. 7E-H). The PZ size showed a strong correlation with ATS extent, mRNA extent, the number of ATS-containing GSCs, or the total number of ATS in the gonad, suggesting that all these parameters can serve as predictors of PZ size. The implications of these relationships and the slopes of their regression models will be further discussed in the discussion.

## DISCUSSION

This work systematically analyzes the molecular and biological consequences of commonly used *C. elegans glp-1*/Notch mutations with human disease relevance at chromosomal (*sygl-1* and *lst-1* ATS), cellular (summed ATS intensities and cellular mRNA), and tissue (distal germline) levels. To this end, we use direct Notch activation readouts, including ATS and mRNA of the major Notch target *sygl-1*, along with complementary functional readouts such as the Notch-responsive GSC pool and PZ size. These analyses reveal

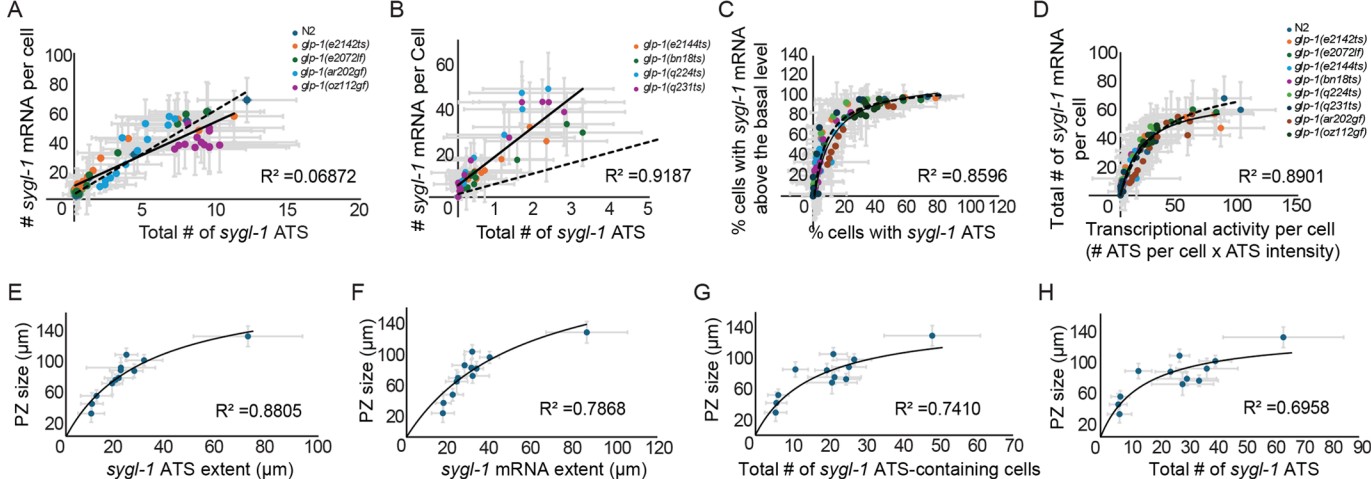

**Fig. 7. Regression analysis of *sygl-1* ATS, mRNA, and germline function in *glp-1* mutants.** (A,B) The number of *sygl-1* mRNA per cell is plotted against the total number of *sygl-1* ATS at the corresponding location in the gonad of *glp-1* NECD (A) and NICD (B) mutants. (A-H) $R^2$ indicates the correlation coefficient of the linear fitting models for all strains (solid black line). Dashed black line indicates the linear model for N2 (wild type). Gray crosses indicate standard deviations for the corresponding axes. (C) The percentage of cells containing *sygl-1* mRNA above the basal level is plotted against the percentage of *sygl-1* ATS at the corresponding location of the gonad. (D) The number of *sygl-1* mRNA per cell is plotted against cellular transcriptional activity, estimated by the product of the number of *sygl-1* ATS per cell and the ATS intensities at the corresponding location in the gonad. (C,D) A hyperbolic model is used for best model fitting. (A-D) *n*=62, 13, 24, 23, 24, 67, 30, 29, and 14 gonads in the order displayed. (E-H) The progenitor zone (PZ) sizes are plotted against *sygl-1* ATS extents (E), mRNA extents (F), the total number of *sygl-1* ATS-containing cells (G), or the total number of *sygl-1* ATS (H) in the corresponding gonads. Temperatures and sample sizes were as follows: N2 (20°C *n*=32 gonads, 22.5°C *n*=20) along with the mutations *glp-1(e2142ts)* (15°C *n*=20, 20°C *n*=13 gonads), *glp-1(e2072lf)* (20°C *n*=23 gonads), *glp-1(e2144ts)* (15°C *n*=20, 22.5°C *n*=19 gonads), *glp-1(bn18ts)* (15°C *n*=20, 22.5°C *n*=24 gonads), *glp-1(q224ts)* (15°C *n*=30 gonads), *glp-1(q231ts)* (15°C *n*=20 gonads), *glp-1(ar202gf)* (15°C *n*=28 gonads).

clear distinctions among several *glp-1* mutations that were previously considered similar just as *lf* or *gf* mutants (Austin and Kimble, 1987; Berry et al., 1997; Kodoyianni et al., 1992). We demonstrate that the effects and severity of *glp-1* mutants depend on their location within GLP-1 and that *glp-1* mutations influencing NICD activity generally have a greater impact on Notch activity at all biological levels than those influencing NICD abundance. In addition, a series of regression analyses reveal strong, simple correlations between Notch-activated *sygl-1* ATS, mRNA levels, and PZ size, regardless of the type of *glp-1* mutation, allowing one parameter to be used to predict others, providing a framework for assessing unexplored *glp-1* mutations. Below we place these insights in context and discuss their broader implications.

### *glp-1* mutations affect Notch activity at the ATS, mRNA, and germline levels, each in a distinct manner

We expected that *glp-1* mutations would alter Notch-induced transcriptional activation (whether the chromosome or cell responds to Notch signaling) as well as its activity (how many RNAs are produced upon activation), which are estimated by *sygl-1* ATS number or its signal intensity (number of nascent RNAs produced), respectively. However, the precise manner in which these mutations modulate transcription across different biological levels remained undetermined. As anticipated, several *glp-1 lf* mutants exhibited a significant reduction in Notch transcriptional activation at the tissue level, as indicated by decreases in the total number of *sygl-1* ATS-containing GSCs, total mRNA numbers, their spatial extents, and the Notch-responsive GSC pool size (Figs 2A,B, 3A,D, and 4). This trend persisted at the cellular level, where reductions in the number of *sygl-1* ATS and mRNAs per cell reflected decreased Notch activity within individual GSCs (Figs 2D,E, 3B,C). However, changes at the chromosomal (ATS) level were comparatively modest, with an average decrease of 20% compared to over 65% reduction at tissue and cellular levels (Fig. 2A,E,F). These results

suggest that while several *glp-1 lf* mutations significantly impair Notch transcriptional activation at broader biological scales, their impact at the chromosomal level is less pronounced. This implies that *glp-1* mutations primarily affect the probability of transcriptional activation rather than the transcriptional capacity (activity) of individual chromosomes.

### *glp-1* NICD mutations have stronger effects on Notch activity than NECD mutations

The *glp-1* mutations analyzed in this study can be categorized into two groups based on the location of the genetic lesion: NICD mutations, which harbor a missense *lf* mutation within the NICD, particularly in the ANK domain, and NECD mutations, where a missense *lf* or *gf* mutation is located in the NECD, specifically within an EGF repeat (Table 1, Figs 1B, 5A). NICD mutations are expected to influence the activity or stability of the NICD, and consequently NTFC function, while having minimal direct impact on the amount of GLP-1/Notch receptor being cleaved upon signal transduction (Greenwald and Kovall, 2018; Kovall et al., 2017). In contrast, NECD mutations are likely to disrupt the interaction between GLP-1 and its ligands, such as LAG-2, thereby primarily affecting NICD abundance rather than its activity, as the cleaved NICD itself remains unmutated (Gordon et al., 2008; Greenwald and Kovall, 2018; Kovall et al., 2017; Nurmahdi et al., 2022). Our findings indicate that only NICD mutations significantly reduce Notch-induced transcriptional activation across all biological levels, albeit to varying extents that correspond to their biological consequences, such as the PZ size, the GSC pool size, and brood size (Berry et al., 1997; Kodoyianni et al., 1992; Lee et al., 2016) (Fig. 2B, Figs S2F, and S3C). In contrast, NECD mutations have a minimal impact on Notch activation and its activity (Figs 2–4). This was further supported by analyses of *sel-12*/Presenilin mutations, which impair GLP-1 cleavage and likely reduce NICD production, as well as *glp-1* NECD *gf* mutations, which lead to autonomous

cleavage and constitutive NICD release (Berry et al., 1997; Jones et al., 2024; Pepper et al., 2003) (Figs 5, 6). Neither sel-12 lf and glp-1 gf mutations substantially altered Notch transcriptional activation or its activity at ATS and cellular mRNA levels, especially at the distal end of the gonad, as well as the GSC pool size in sel-12 mutants (Figs 5E-I, and 6). These findings suggest that NICD abundance plays a minimal role in regulating Notch activation and activity, whereas the activity or stability of NICD has a more profound impact, possibly through modulating NTFC function. sel-12 has a paralog, hop-1, which is partially redundant for Notch activation (Li and Greenwald, 1997). Further studies examining hop-1 mutants will help clarify the role of presenilin and NICD abundance in regulating Notch-induced transcription. Additionally, the EGF mutant glp-1(oz25), which carries a missense mutation in the NECD, exhibits a glp-1 null phenotype unlike the NECD mutants examined in this study (Kodoyianni et al., 1992). Also, glp-1(e2142) and glp-1(e2072), despite their minimal effects in the germline, exhibit clear defects in embryonic development (Kodoyianni et al., 1992). Further analyses of additional NECD mutants will be important for understanding its precise role in Notch transcriptional regulation.

### The graded Notch activation pattern is largely maintained in most glp-1 mutants

Surprisingly, despite the substantial tissue- and cellular-level changes in Notch activation and its activity observed in glp-1 lf mutants, the majority of mutations preserved the graded sygl-1 transcriptional pattern in the distal gonad, although the gradient was not as steep as in wild type (Fig. 4A). This gradient also persisted in glp-1 gf and sel-12 lf mutants (Figs 5K-M, 6F,G), suggesting that neither NICD abundance nor its activity/stability plays a critical role in establishing the graded Notch activation pattern. However, one mutation, glp-1(q231), abolished the graded pattern for both sygl-1 ATS and mRNAs, indicating that specific transcription factors (TFs), mediators, or co-transcriptional activators, such as LAG-1/CSL and LAG-3/MAML (a.k.a. SEL-8), may interact with the residue where glp-1(q231) is located to regulate the graded Notch response. These factors might be expressed in a graded manner, similar to the sygl-1 ATS gradient in the distal gonad, which could be further investigated through mRNA or protein localization studies. Alternatively, upstream of GLP-1 cleavage, Notch ligands such as LAG-2 or APX-1 may be asymmetrically localized on the DTC, allowing GSCs closer to the DTC to receive stronger Notch signaling and consequently exhibit higher transcriptional activation levels.

### Revealing a simple, direct correlation between ATS, mRNA, and germline function

Notably, all glp-1 lf and gf mutations analyzed in this study exhibited a strong linear correlation between sygl-1 ATS count and mRNA count per cell, indicating that overall cellular transcriptional activation is directly reflected in mRNA products (Fig. 7A,B, Fig. S10). This linear relationship suggests minimal regulatory steps between Notch-induced nascent RNA generation and cytoplasmic mRNA accumulation. However, the regression slopes were different between mutations affecting NICD abundance (EGF mutations and gf mutations) and those impacting NICD activity or stability (ANK mutations). Mutations affecting NICD abundance exhibited slopes similar to wild type, whereas ANK mutations displayed a steeper slope. This suggests that reduced NICD activity or stability may also impair mRNA degradation, possibly through unidentified Notch targets involved in mRNA processing or by altering cellular conditions. This strong correlation was also observed at the tissue

level, where the percentage of sygl-1 ATS-containing cells closely matched the percentage of sygl-1 mRNA-containing cells at each cell row along the germline, following a hyperbolic relationship with saturation around 50% (Fig. 7C). This indicates that when at least 50% of GSCs at a given cell row exhibit sygl-1 ATS, nearly all cells at that position contain sygl-1 mRNAs above the basal level, effectively designating them as GSCs. This finding aligns with the spatial pattern of sygl-1 mRNAs, where the first 10-15 µm from the distal end of the gonad does not exhibit a steep gradient, unlike the ATS pattern in wild type and EGF mutants, as at least 50% of GSCs in this region contain ATS. These relationships were further supported by a similarly strong correlation between overall cellular Notch activity and sygl-1 mRNA count per cell, regardless of glp-1 mutation type (Fig. 7D). Notably, we observed a moderate to strong correlation between these readouts and PZ size, a well-characterized and commonly used measure of gametogenesis capacity, suggesting that these Notch readouts can serve as germline function indicators.

## MATERIALS AND METHODS
### Nematode strains used in this study:

| Strain name | Genotype |
| --- | --- |
| N2 | Wild-type strain of C. elegans |
| CF1903 | glp-1(e2144ts) III |
| DG2389 | glp-1(bn18ts) III |
| JK509 | glp-1(q231ts) III |
| JK4605 | glp-1(q224ts) III |
| JK1505 | unc-32(e189) glp-1(e2072lf) III/ eT1(III; V) |
| JK289 | glp-1(e2142ts) III |
| BS860 | unc-32(e189) glp-1(oz112gf)/ dpy-19(e1259) glp-1(q172) |
| BS3164 | unc-32(e189) glp-1(ar202gf) III |

### Nematode culture

All strains were maintained at 15°C as described (Austin and Kimble, 1987; Kodoyianni et al., 1992; Pepper et al., 2003), except N2, glp-1(e2072lf) and glp-1(oz112gf), which were maintained at 20°C (Berry et al., 1997; Brenner, 1974). The wild type was N2 Bristol. Mutations were as follows: glp-1(e2144ts) III (Kodoyianni et al., 1992; Lee et al., 2016), glp-1(bn18ts) III (Fox and Schedl, 2015; Kodoyianni et al., 1992), glp-1(q231ts) III (Kodoyianni et al., 1992), glp-1(q224ts) III (Kodoyianni et al., 1992; Lee et al., 2016), unc-32(e189) glp-1(e2072lf) III/ eT1(III; V) (Kodoyianni et al., 1992), glp-1(e2142ts) III (Kodoyianni et al., 1992), unc-32(e189) glp-1(oz112 gf)/ dpy-19(e1259) glp-1(q172) (Berry et al., 1997), and unc-32(e189) glp-1(ar202) III (Pepper et al., 2003). For the smFISH experiments, all strains were synchronized and cultured on OP50 seeded NGM plates until the YA stage (24 h post mid L4 in 20°C and 36 h post mid L4 in 15°C).

### Single-molecule RNA fluorescence in situ hybridization (smFISH)

smFISH for sygl-1 and let-858 were performed as previously described (Lee et al., 2016, 2017; Urman et al., 2024). Synchronized L1 larvae (Kershner et al., 2014) were grown on OP50 until YA stage within their respective temperatures. The synchronized YA C. elegans of each of the strains were washed off plates with 2-3 ml non-RNase free PBST and were collected on the 60 mm Petri dish cover. An additional 2-3 ml of non-RNase free PBST was added, and the worms dissected to extrude the gonads in PBST with 0.25 mM levamisole added. The following reagents were prepared: PBST (RNase-free 1X PBS+0.1% Tween-20), permeabilization buffer (RNase-free 1X PBS+0.1% Triton X-100), and RNase-free 70% ethanol. Samples were then fixed with 3.7% formaldehyde in 1X PBS with 0.1% Tween-20 at room temperature (RT) for 15-45 min, with gentle agitation. Samples were pelleted at 2000 RPM for 30 s unless noted otherwise. After fixation, samples were permeabilized with PBST containing 0.1% Triton X-100 for

10 min at RT with rotation, washed twice with PBST, resuspended in 70% ethanol, and stored overnight at 4°C.

Custom Stellaris FISH probes (Biosearch Technologies, Inc., Petaluma, CA, USA) were designed against the exon and intron regions of *sygl-1* and the intron regions of *let-858* as follows. The probe set designed for *sygl-1* exon regions includes 48 unique oligonucleotides labeled with TAMRA. The probe set designed for *sygl-1* intron regions includes 48 unique oligonucleotides tagged with Quasar 670. The probe set designed for *let-858* intron regions includes 48 unique oligonucleotides labeled with TAMRA. The dried probe mix (5 nmol) were dissolved to 40 µl of RNase-free TE buffer (10 mM Tris-HCl, 1 mM EDTA, pH 8.0) to create a probe stock of 125 µM. The probe stock was then diluted down in RNase-free TE buffer 1:20 (6.25 µM).

Ethanol was removed and samples incubated in wash buffer (2X SSC, 10% deionized formamide in nuclease-free water) for 5 min at RT. Gonads were hybridized with 1 µl of each of the *sygl-1* probes (6.25 µM) or *let-858* probes (6.25 µM) in hybridization solution (228 mM Dextran sulfate, 2X SSC, 10% deionized formamide in nuclease-free water) 24 h at 37°C with rotation. After probe addition, samples were kept in the dark for all incubations and washes. Samples were rinsed once with wash buffer at RT, then incubated in wash buffer for 30 min at room temperature with rotation. The DNA was then labeled by incubation in smFISH wash buffer containing 1 mg/ml diamidinophenylindole (DAPI) for 30 min at RT followed by two short washes with smFISH wash Buffer. Finally, samples were resuspended in 10-12 µl Antifade Prolong Gold mounting medium (Life Technologies Corporation, Carlsbad, CA, USA) and mounted on glass slides.

### Microscopy setup and image acquisition

Gonads were imaged as previously described (Urman et al., 2024), using a Leica DMi8 Widefield Microscope, that is equipped with a Leica HC PL APO 63×/1.40-0.60 NA oil immersion objective, LED8 fluorescence illuminator, and THUNDER Imager with exceptional computational clearing methods to remove excessive background. All gonads were imaged completely (depth >15 µm) with a Z-step size of 0.3 µm using the Leica Application Suite X (LAS X) acquisition software (Leica Microsystems Inc., Buffalo Grove, IL, USA). All imaging was done with LED8 light sources. Channels were sequentially imaged in decreasing wavelengths to avoid bleed-through and prevent any photobleaching from occurring. The *sygl-1* exon probe (TAMRA) and the *let-858* intron probe (TAMRA) were excited at 555 nm (40%) and the signals were acquired at 540-640 nm (gain was set to high well capacity) with an exposure time of 250 ms. The *sygl-1* intron probe (Quasar 670) was excited at 635 nm (40%) and the signal was acquired at 625-775 nm (gain was set to high well capacity) with an exposure time of 250 ms. DAPI was excited at 390 nm (10% illumination), and signal was acquired at 400-480 nm (gain high well capacity) with an exposure time of 50 ms.

### PZ extents

The PZ is a segment of the germline at the distal end that contains germline stem cells, their proliferating progeny, and cells in meiotic S phase (Tolkin and Hubbard, 2021). The end of the PZ was defined as a cell row with more than one cell exhibiting a crescent-shaped morphology (Byrd et al., 2014; Crittenden et al., 2006; Fausett et al., 2023; Gordon, 2020; Roy et al., 2016; Urman et al., 2024). In this study, the PZ extent was measured using a segmented line on ImageJ from the most distal end of the gonad to the end of the progenitor zone.

### Notch *sygl-1* ATS and mRNA extents

For *sygl-1* ATS extents, the extent was assessed by measuring the distance from the center of the first ATS at the distal most end of the germline to the last ATS within the germline. For *sygl-1* mRNA extents, the extent was assessed by measuring the distance from the first mRNA at the distal most end of the germline to the end of the mRNA rich region (<5 mRNA per cell) within the germline.

### Image processing using the custom-made MATLAB codes

All processes were implemented and automated using modified MATLAB (v2.0) codes similar to the source code developed in our previous work (Crittenden et al., 2019; Lee et al., 2016; Lynch et al., 2022; Urman et al.,

2024) with certain modifications to optimize the source code for use with Widefield microscopy images of mutant strains. The MATLAB code modifications made for this study were particularly in RNA detection portions of the source code. For ATS detection, the signal-to-local background ratio method was implemented as previously described criteria (Urman et al., 2024). Modifications in mRNA detection was set up similarly to ATS detection described above with the exception that it was based on *sygl-1* exon-specific probes, with certain modifications in the 'DetectRNAexon' function to improve detection of mRNAs in close proximity and detect these mRNA via the exon channel, as reported previously (Crittenden et al., 2019; Lynch et al., 2022; Urman et al., 2024). Our previous work optimized these codes that the ATS intensities were measured from the exon probes and were used in the direct comparison between ATS and mRNA counts and their intensities with the mean cytoplasmic mRNA intensities for all gonads was set to 1 arbitrary unit (a.u.) (Urman et al., 2024). The number of mRNAs within the germ cells were estimated using the germ cell boundary, which was established using a 3D Voronoi diagram with the size or radius of each germ cell was restricted to 3 µm from the nucleus center (Ledoux, 2007; Yan et al., 2010). After the analysis was completed, MATLAB and GraphPad Prism were used to visualize the data generated and conduct statistical tests. ANOVA and *t*-tests were performed to compare data if the data set met the requirements for parametric statistical analysis through normality tests (Anderson-Darling Normality test) and Kolmogorov–Smirnov (KS) test (a nonparametric version of the *t*-test) was performed if the data set did not satisfy the requirements for parametric analysis.

### Acknowledgements

We are thankful for the resources provided by the Molecular Biology Core Facility in the Life Sciences Research Building and the RNA Institute at the University at Albany.

### Competing interests

The authors declare no competing or financial interests.

### Author contributions

Conceptualization: C.L., N.S.J.; Data curation: N.S.J.; Formal analysis: C.L., N.S.J., M.A.U., M.G.M.; Funding acquisition: C.L.; Investigation: N.S.J., M.A.U., M.G.M.; Methodology: N.S.J.; Project administration: C.L.; Resources: C.L.; Supervision: C.L.; Validation: C.L., N.S.J., M.A.U.; Visualization: N.S.J.; Writing – original draft: N.S.J.; Writing – review & editing: C.L., M.A.U., M.G.M., V.G.

### Funding

This work was funded by the University at Albany (FRAP-A Award 1189585-1-97969). Open Access funding provided by University at Albany. Deposited in PMC for immediate release.

### Data and resource availability

All relevant data and details of resources can be found within the article and its supplementary information.

### Peer review history

The peer review history is available online at https://journals.biologists.com/bio/lookup/doi/10.1242/bio.062008.reviewer-comments.pdf

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
