## [Peer Review File · Biology Open]

Genetic mutations in GLP-1/Notch pathway reveal distinct mechanisms of Notch signaling in germline stem cell regulation

Nimmy Sara John, Michelle Alexandra Urman, Mahasin Gulnaaz Mehmood, Vanessa Gentile and ChangHwan Lee

DOI: 10.1242/bio.062008

Editor: Alissa Armstrong

Review timeline

Original submission:	2 April 2025
Editorial decision:	11 April 2025
First revision received:	18 August 2025
Editorial decision:	28 August 2025
Second revision received:	5 December 2025
Accepted:	11 December 2025

Original submission

First decision letter

MS ID#: bio.062008

MS Title: Genetic mutations in GLP-1/Notch pathway reveal distinct mechanisms of Notch signaling in germline stem cell regulation

Authors: ChangHwan Lee, Nimmy Sara John, Michelle Alexandra Urman, Mahasin Gulnaaz Mehmood and Vanessa Gentile

I have now reached a decision on the above manuscript.

The reviewer reports are shown at the bottom of this email or can be accessed, together with a copy of this decision letter, by going to:

As you will see, the reviewers raised a number of substantial criticisms that prevent me from accepting the paper at this stage. Given Reviewer 1's major concerns about the experimental approach, it is crucial that these be adequately addressed in a revision.

They suggest, however, that a revised version might prove acceptable, if you can address their concerns. If you think that you can deal satisfactorily with the criticisms on revision, I would be pleased to see a revised manuscript. We would then return it to the reviewers.

At this stage, we also ask you to ensure your manuscript complies with our formatting guidelines. Provided you are able to fully address the referees' comments, we are positive about publication of your paper (we accept over 95% of revision submissions) and therefore hope you won't mind any extra work involved in reformatting your manuscript at this point.

Please ensure that you clearly highlight all changes made in the revised manuscript. Please avoid using 'Tracked changes' in Word files as these are lost in PDF conversion.

I should be grateful if you would also provide a point-by-point response detailing how you have dealt with the points raised by the reviewers in the 'Response to Reviewers' box. Please attend to

all of the reviewers' comments. If you do not agree with any of their criticisms or suggestions please explain clearly why this is so.

Reviewer 1

Comments for the author

Despite the characterization of several mutant strains targeting the Notch signalling pathway, there are concerns regarding the choice of controls and the strength of the conclusions drawn from the experimental data.

Major comments:

1) While the authors state that Notch signaling does not significantly differ across control strains incubated at various temperatures, the representative images in Figure S1B and quantification in Figure S1D suggest otherwise. Specifically, there is an apparent expansion in the proportion of cells with active transcription sites (ATS), spanning over 20 μm and including six bars of quantified cells instead of the typical five in controls. This discrepancy is notable, especially given that the same analysis framework is later used to compare across mutant strains.

Additionally, Figure S2 shows differences between control and mutant strains even at permissive temperatures, suggesting that temperature may influence Notch signalling independently of mutation status. These observations raise concerns about using 20°C as a universal control temperature, particularly for strains with temperature-sensitive phenotypes. The authors should re-analyze the data using temperature-matched controls for each strain to ensure accurate interpretation.

2) The authors rely on *sygl-1* expression as a direct indicator of germline stem cell (GSC) number or pool size. However, since *sygl-1* is a downstream target of Notch signalling, its expression is inherently affected in the mutants being studied. This approach introduces a circular logic and could confound the interpretation of GSC dynamics. The authors should include an additional GSC marker independent of the Notch pathway to more accurately assess GSC number. If such staining is not feasible, the authors should refrain from making definitive claims about GSC pool size changes.

3) The current discussion section lacks a comprehensive analysis of how the findings integrate with existing knowledge of Notch mutations, signalling mechanisms, and functional outcomes, in particular focusing in the differences between NICD and NECD mutations. The discussion should be expanded to place the findings in the broader context of Notch signalling research, with emphasis on domain-specific effects and potential mechanisms driving the observed phenotypes.

Minor comments:

* The separation of figures from the analysis makes the manuscript difficult to follow. This is especially evident for Figure 1C, a central figure that is only briefly mentioned without proper description. I suggest to reorganize the figure presentation by grouping mutants into relevant categories (e.g., EGF vs. ANK domain mutants) and including both representative images and corresponding quantification together.

* Include the permissive and restrictive temperatures for each mutant in the mutant strain table to aid interpretation.

* Quantitative results would be more informative if the manuscript included percent reductions observed in various analyses. This will help readers evaluate the biological impact of the mutations on Notch signalling.

* Maintain consistency in graph formats throughout the manuscript. Some figures use bar plots, while others use violin plots (e.g., Figures 5, 6, and S8). Standardize the format and ensure violin plots include all individual data points.

* The authors describe differences across "chromosomal, cellular, and tissue levels," but these terms do not accurately reflect the study's focus. Replace "chromosomal" and "tissue" with more precise terms such as "nuclear ATS" and "germline" to better represent the analysis levels.

* There is no detailed explanation of how the proliferative zone (PZ) size is determined, nor are images provided showing PZ size variation across strains.

* Figure S3F is referenced but appears to be missing. Please verify and correct this.

Reviewer 2

Comments for the author

This interesting piece by John et al. explores the transcriptional defects caused by a range of *C. elegans* *glp-1*/Notch mutations affecting either its NECD or NICD. Single-molecule FISH analyses on a Notch direct transcriptional target (*sygl-1*) are used to mark both nascent (unspliced, corresponding to Active Transcriptional Sites, ATS) and total (spliced and unspliced) RNAs in extruded and fixed germlines. Noteworthy conclusions are reached including that NECD(lf) mutations appear (though see below) to cause milder transcriptional defects than NICD(lf) mutations. Results therefore strengthen the differences between these two distinct classes of *glp-1* mutants, as they were initially proposed by Kodoyianni et al, 1992. More interestingly, a strong correlation that holds true across all mutants demonstrates that ATS analyses reliably predict global transcription and vice-versa.

Major

1- In the abstract and elsewhere, authors claim: "results reveal that NICD mutants reduce Notch transcriptional activation... whereas NECD mutations have minimal effects...". However this is incorrect and a bold overstatement. First, this only concerns NECD(lf) mutations as NECD(gf) have significant effects. Second, as only two NECD(lf) mutations were analysed it is difficult to argue that this represents a general phenomenon. Additional NECD(lf) mutations would need to be analysed. Moreover, the *oz25* allele is a single amino acid substitution in an NECD EGF-like motif but causes a null-like phenotype (Kodoyianni et al, 1992). This allele would likely severely affect *sygl-1* transcription. Therefore, authors cannot claim that NECD mutations have little effects on Notch transcriptional activation.

2- Authors use *sel-12* alleles to claim that decreasing the amount of NICD produced does not affect *sygl-1* transcriptional activation but it is unclear that the *sel-12* alleles used do significantly reduce the amount of NICD produced. Indeed, *sel-12* is partially redundant with another presenilin gene, *hop-1*, and it is unclear that *sel-12* mutants do decrease *glp-1* cleavage on their own since these single mutants do not have an obvious germline phenotype. NICD internalization should be measured in these backgrounds using a tagged *glp-1* (NICD) allele.

Minor

1- The two *gf* alleles should be indicated in Fig. 1B.

2- The two NECD(lf) alleles studied have a mild germline defect, but a stronger effect on embryonic development. It would be important to consider this when interpreting results.

3- The use of partially restrictive conditions (22.5C instead of 25C) may unequally affect the severity of the defects. It is unclear whether the conclusions on the two weaker NECD(lf) alleles would hold true at 25C? This limitation should be pointed out.

4- Please indicate the temperatures used on all figures. Also explain the rationale for using 22.5C instead of 25C for temperature sensitive alleles.

5- The flow of thoughts during the first two paragraphs of the results section could be improved (smFISH technique, temperature issues, mutant selection, results). It would perhaps be more logical to begin the results by explaining the question to be addressed, the rationale for choosing the selected *glp-1* (lf) mutations, followed by temperature issues, smFISH technique, then results.

6- The use of the expression "clinically relevant" is somewhat misleading as the *C. elegans* *glp-1* mutations studied do not directly correspond to pathology-causing human mutations. "Notch mutations that are relevant to human diseases" is more appropriate.

7- Fig. 2C, change the y-axis title for "# ATS/um"?

8- The statement "let-858, a Notch-independent gene constitutively expressed in the germline" needs a reference.

9- The logic behind the statement: "the average nuclear size in the distal gonad remained unchanged across wild-type and *glp-1* mutants, indicating that these mutations are unlikely to affect cell cycle progression or its rate." is not clear -please develop the rationale - and such a statement also requires a reference.

10- Line 182: Replace "GSCs" by "germ cells".

11- "analyses reveal clear distinctions among *glp-1* mutations that were previously considered similar" this is ignoring the Kodoyianni paper.

Reviewer's Responses to Questions

Experimental quality

Does each figure have the proper controls?

If 'No', please indicate reasons in Comments for Author box below.

Reviewer #1:

- No

Reviewer #2:

- Yes

Were the data analyzed using appropriate statistical tests?

If 'No', please indicate reasons in Comments for Author box below.

Reviewer #1:

- Yes

Reviewer #2:

- Yes

Reproducibility

Were experiments performed using adequate number of biological replicates?

If 'No', please indicate reasons in Comments for Author box below.

Reviewer #1:

- Yes

Reviewer #2:

- Yes

Does the methods section provide sufficient detail to permit reproducibility?

If 'No', please indicate reasons in Comments for Author box below.

Reviewer #1:

- No

Reviewer #2:

- Yes

Completeness

Are the manuscript's conclusions supported by the data?

If 'No', please indicate reasons in Comments for Author box below.

Reviewer #1:

- No

Reviewer #2:

- No
-

Scholarship

Do the authors cite and discuss the merits of data that would argue for and against their conclusion?

If 'No', please indicate reasons in Comments for Author box below.

Reviewer #1:

- Yes

Reviewer #2:

- No

Does the manuscript title & abstract accurately reflect the contents of the manuscript, without hyperbole?

If 'No', please indicate reasons in Comments for Author box below.

Reviewer #1:

- Yes

Reviewer #2:

- Yes

First revisionAuthor response to reviewers' comments**Comments from the Reviewers:**

Reviewer 1: Despite the characterization of several mutant strains targeting the Notch signalling pathway, there are concerns regarding the choice of controls and the strength of the conclusions drawn from the experimental data.

Major comments:

1) While the authors state that Notch signaling does not significantly differ across control strains incubated at various temperatures, the representative images in Figure S1B and quantification in Figure S1D suggest otherwise. Specifically, there is an apparent expansion in the proportion of cells with active transcription sites (ATS), spanning over 20 μm and including six bars of quantified cells instead of the typical five in controls. This discrepancy is notable, especially given that the same analysis framework is later used to compare across mutant strains. Additionally, Figure S2 shows differences between control and mutant strains even at permissive temperatures, suggesting that temperature may influence Notch signalling independently of mutation status. These observations raise concerns about using 20°C as a universal control temperature, particularly for strains with temperature-sensitive phenotypes. The authors should re-analyze the data using temperature-matched controls for each strain to ensure accurate interpretation.

We appreciate the reviewer's thoughtful feedback. In response, we have added N2 (wild-type) controls at 15, 20, and 22.5°C to all relevant plots to enable direct comparison with *glp-1* mutants. All statistical analyses have been redone using the corresponding temperature-matched controls (see revised Fig. 2-5 and relevant Supplementary Figures). These updates, however, did not alter our statistical outcomes or the overall conclusions of the study; therefore, changes to the Results and Discussion sections were minimal. Additionally, we have updated the images in Fig. S1B to more accurately reflect the quantitative analysis presented in Fig. S1D. As the reviewer rightly noted, *sygl-1* ATS or mRNA levels and spatial patterns slightly change in certain *glp-1* mutants even at permissive temperatures. This is consistent with previous reports showing that these mutants exhibit mild physiological and behavioral defects under permissive conditions (e.g., Austin et al. 1987). However, our regression analyses demonstrate that the relationship among *sygl-1* ATS, mRNA, and germline function remain consistent across temperatures (Fig. 7 and S9-S12). Together,

these results suggest that temperature has a minimal impact on Notch pathway activation in our experimental context. We have revised the Discussion to clarify this point.

2) The authors rely on *sygl-1* expression as a direct indicator of germline stem cell (GSC) number or pool size. However, since *sygl-1* is a downstream target of Notch signalling, its expression is inherently affected in the mutants being studied. This approach introduces a circular logic and could confound the interpretation of GSC dynamics. The authors should include an additional GSC marker independent of the Notch pathway to more accurately assess GSC number. If such staining is not feasible, the authors should refrain from making definitive claims about GSC pool size changes.

Besides Notch-induced *sygl-1* ATS readout, the only other established and validated GSC marker is to identify mitotic cells that can be arrested at M phase using the *emb-30* assay (Cinquin et al. 2010 Curr Biol). This assay, however, requires an *emb-30* null temperature-sensitive mutant background, which disrupts GSC biology as well as cell cycle progression, germline morphology, and brood size, leaving it as a less ideal assay, especially in mutant backgrounds that already alter germline function, such as *glp-1* mutants. Multiple attempts to genetically cross our *glp-1* mutants with *emb-30* mutants resulted in infertility, inviability, or several germline morphological defects. Thus, we have revised our manuscript to refer exclusively to the Notch-responsive GSC pool examined. As a complementary readout to *sygl-1*, we added new data analyzing a second Notch target, *lst-1*, in *glp-1* mutant backgrounds (Fig. S13). Although both *sygl-1* and *lst-1* are activated by Notch activation in the germline, their transcriptional activation and spatial patterns are independent, making *lst-1* a suitable alternative marker for Notch activity. Our results with *lst-1* in *glp-1* mutant backgrounds showed essentially the same results, supporting our conclusions.

3) The current discussion section lacks a comprehensive analysis of how the findings integrate with existing knowledge of Notch mutations, signalling mechanisms, and functional outcomes, in particular focusing in the differences between NICD and NECD mutations. The discussion should be expanded to place the findings in the broader context of Notch signalling research, with emphasis on domain-specific effects and potential mechanisms driving the observed phenotypes.

We appreciate the reviewer's suggestion. We have revised our Discussion to incorporate the aspects the reviewer indicated.

Minor comments:

* The separation of figures from the analysis makes the manuscript difficult to follow. This is especially evident for Figure 1C, a central figure that is only briefly mentioned without proper description. I suggest to reorganize the figure presentation by grouping mutants into relevant categories (e.g., EGF vs. ANK domain mutants) and including both representative images and corresponding quantification together.

To improve clarity in the figures, we added labels indicating mutant categories based on their domain location (e.g., EGF, ANK) and expanded the accompanying text with additional details.

* Include the permissive and restrictive temperatures for each mutant in the mutant strain table to aid interpretation.

We added an additional column indicating the temperatures to Table 1.

* Quantitative results would be more informative if the manuscript included percent reductions observed in various analyses. This will help readers evaluate the biological impact of the mutations on Notch signalling.

We added percent reductions where applicable in the text.

* Maintain consistency in graph formats throughout the manuscript. Some figures use bar plots, while others use violin plots (e.g., Figures 5, 6, and S8). Standardize the format and ensure violin

plots include all individual data points.

All bar plots have been replaced with violin plots for consistency. Many datasets contain thousands to tens of thousands of data points, effectively filling the entire violin shapes. We are open to further modifying plot formats if requested by the reviewers.

* The authors describe differences across "chromosomal, cellular, and tissue levels," but these terms do not accurately reflect the study's focus. Replace "chromosomal" and "tissue" with more precise terms such as "nuclear ATS" and "germline" to better represent the analysis levels.

We have revised our manuscript to address the reviewer's concern.

* There is no detailed explanation of how the proliferative zone (PZ) size is determined, nor are images provided showing PZ size variation across strains.

We have added or expanded explanations of how the measurements were performed systematically in figure legends, Results, and Methods sections, and have included relevant references.

* Figure S3F is referenced but appears to be missing. Please verify and correct this. We have corrected it.

Reviewer 2: This interesting piece by John et al. explores the transcriptional defects caused by a range of *C. elegans* *glp-1*/Notch mutations affecting either its NECD or NICD. Single-molecule FISH analyses on a Notch direct transcriptional target (*sygl-1*) are used to mark both nascent (unspliced, corresponding to Active Transcriptional Sites, ATS) and total (spliced and unspliced) RNAs in extruded and fixed germlines. Noteworthy conclusions are reached including that NECD(lf) mutations appear (though see below) to cause milder transcriptional defects than NICD(lf) mutations. Results therefore strengthen the differences between these two distinct classes of *glp-1* mutants, as they were initially proposed by Kodoyianni et al, 1992. More interestingly, a strong correlation that holds true across all mutants demonstrates that ATS analyses reliably predict global transcription and vice-versa.

Major

1- In the abstract and elsewhere, authors claim: "results reveal that NICD mutants reduce Notch transcriptional activation... whereas NECD mutations have minimal effects...". However this is incorrect and a bold overstatement. First, this only concerns NECD(lf) mutations as NECD(gf) have significant effects. Second, as only two NECD(lf) mutations were analysed it is difficult to argue that this represents a general phenomenon. Additional NECD(lf) mutations would need to be analysed. Moreover, the *oz25* allele is a single amino acid substitution in an NECD EGF-like motif but causes a null-like phenotype (Kodoyianni et al, 1992). This allele would likely severely affect *sygl-1* transcription. Therefore, authors cannot claim that NECD mutations have little effects on Notch transcriptional activation.

We agree with the reviewer's critique. Although all widely used NECD mutants examined in our study (2 EGF, 1 LNR, and 1 HD) showed minimal effects on Notch-induced *sygl-1* transcription at the chromosomal and cellular levels, two *gf* NECD mutants (LNR and HD) altered Notch activation at the tissue level, expanding Notch-responsive GSC pool by changing the spatial pattern of *sygl-1* transcription. We have revised the manuscript to clarify these distinct effects of different mutations and updated the Discussion to avoid over-generalization and over-simplification in our NICD and NECD mutant analyses and conclusions.

2- Authors use *sel-12* alleles to claim that decreasing the amount of NICD produced does not affect *sygl-1* transcriptional activation but it is unclear that the *sel-12* alleles used do significantly reduce the amount of NICD produced. Indeed, *sel-12* is partially redundant with another presenilin gene, *hop-1*, and it is unclear that *sel-12* mutants do decrease *glp-1* cleavage on their own since these single mutants do not have an obvious germline phenotype. NICD internalization should be measured in these backgrounds using a tagged *glp-1*(NICD) allele.

Quantifying nuclear NICD is one of our lab's current research goals, but it remains extremely challenging. Multiple attempts by our group and others (e.g., Kimble Lab, Cinquin Lab, Bray Lab) have so far not been successful in achieving this in developing and adult *C. elegans* germlines and *Drosophila*. We recently generated new CRISPR or transgenic strains expressing epitope-tagged *glp-1* (e.g., 4xV5, myc, 6His) and tested available reagents (Sorensen et al. 2022) to quantify internalized NICD, but none produced quantifiable, clear signal, particularly in *glp-1* mutant background. We have revised the Discussion to acknowledge the reviewer's concern and outline future experiments to improve this analysis.

Minor

1- The two *gf* alleles should be indicated in Fig. 1B.

We have revised Fig. 1B as suggested.

2- The two NECD(lf) alleles studied have a mild germline defect, but a stronger effect on embryonic development. It would be important to consider this when interpreting results.

We have revised our manuscript, particularly the Discussion, to address this concern.

3- The use of partially restrictive conditions (22.5C instead of 25C) may unequally affect the severity of the defects. It is unclear whether the conclusions on the two weaker NECD(lf) alleles would hold true at 25C? This limitation should be pointed out.

The two NECD mutants behave as *glp-1* null alleles at 25C, leading to germline loss at 25°C, making them unquantifiable at that restrictive temperature. We have revised our manuscript to clarify this point.

4- Please indicate the temperatures used on all figures. Also explain the rationale for using 22.5C instead of 25C for temperature sensitive alleles.

We have revised the relevant figures and texts as suggested.

5- The flow of thoughts during the first two paragraphs of the results section could be improved (smFISH technique, temperature issues, mutant selection, results). It would perhaps be more logical to begin the results by explaining the question to be addressed, the rationale for choosing the selected *glp-1* (lf) mutations, followed by temperature issues, smFISH technique, then results.

We have revised the manuscript as suggested.

6- The use of the expression "clinically relevant" is somewhat misleading as the *C. elegans glp-1* mutations studied do not directly correspond to pathology-causing human mutations. "Notch mutations that are relevant to human diseases" is more appropriate.

We have revised the manuscript as suggested.

7- Fig. 2C, change the y-axis title for "# ATS/um"?

We have corrected the figure labels as suggested.

8- The statement "let-858, a Notch-independent gene constitutively expressed in the germline" needs a reference.

We have added a relevant reference.

9- The logic behind the statement: "the average nuclear size in the distal gonad remained unchanged across wild-type and *glp-1* mutants, indicating that these mutations are unlikely to affect cell cycle progression or its rate." is not clear -please develop the rationale - and such a statement also requires a reference.

We have added the rationale to the text along with a reference.

10- Line 182: Replace "GSCs" by "germ cells".

We have revised the manuscript.

11- "analyses reveal clear distinctions among glp-1 mutations that were previously considered similar" this is ignoring the Kodoyianni paper.

We have revised the text to discuss the findings in Kodoyianni paper with proper citations.

Second decision letter

MS ID#: bio.062008R1

MS Title: Genetic mutations in GLP-1/Notch pathway reveal distinct mechanisms of Notch signaling in germline stem cell regulation

Authors: ChangHwan Lee, Nimmy Sara John, Michelle Alexandra Urman, Mahasin Gulnaaz Mehmood and Vanessa Gentile

I have now reached a decision on the above manuscript.

The reviewer reports are shown at the bottom of this email or can be accessed, together with a copy of this decision letter, by going to:

The revised version of your manuscript was reassessed by the original reviewers. As you will see, the reviewers generally agreed that many of their concerns/issues raised were appropriately addressed; however, some critical points were not. Each reviewer has provided comments on the revisions and have clarified original issues that were not appropriately address. In order to consider this manuscript for publication, several amendments to your manuscript are required. I hope that you will be able to carry these out, because we would like to be able to accept your paper.

Reviewer 1

Comments for the author

The authors have satisfactorily addressed all of my major comments. I now only have a few minor points concerning the interpretation of the data in the text that should be revised or discussed. If these changes are incorporated, I do not need to review the manuscript again.

1. The authors replaced some instances of "GSCs" with "Notch-responsive GSCs" as suggested (lines 142-143), but this change was not implemented consistently throughout the manuscript. Please replace "GSCs" with "Notch-responsive GSCs" in all cases where GSCs are identified by using Sygl-1 staining.
2. I disagree with the conclusion that NICD mutations play a "more critical role" (line 161) or "have a greater impact on Notch activity" than NECD mutations, which are described as having a "minimal impact." Under restrictive conditions, both NICD and NECD mutants result in infertility. A more accurate conclusion would be that NICD mutations, under partially restrictive conditions, produce a stronger *lf* effect than NECD. This suggests that the NICD mutations tested are more temperature-sensitive than the NECD mutations. This point could be reframed accordingly, and it may be worth noting as an interesting avenue for future research (line 406).
3. I disagree with the statement that *glp-1(gf)* mutations do not alter Notch transcriptional activation, spatial pattern, or activity at ATS, mRNA levels, and GSC pool size (line 393-394). Figure 5 shows evidence of Notch overactivation, with increased ATS and mRNA extents (Fig. 5A, C, D). Moreover, the manuscript states that the gradient persists in *glp-1(gf)* mutants (lines 410-412), but this is not correct. Specifically, *oz112* does not display a gradient, as shown in Fig. 5B and K-L.

4. In lines 424-425, the authors claim that the mutations show a strong linear correlation between ATS and mRNA counts per cell. However, in Fig. 7A the R^2 is not strong, and the data distribution could also be consistent with a hyperbolic relationship.

Reviewer 2

Comments for the author

This is an improved resubmission. However, it seems like the authors did not give enough importance to one of my comments, so I'll have to be more specific about it. There is indeed one main outstanding issue that I am still uncomfortable with as it would be misleading for someone just reading the abstract. There are a few other outstanding details to fix. Everything can be addressed editorially at this point.

Outstanding issues to address:

- 1- The statement in the abstract that "NECD mutations have minimal effects across all biological levels" is not true and has to be modified or deleted. Authors themselves agreed that *gf* NECD mutants do have significant effects. In addition, they say that one of the two NECD *lf* mutant that they used in the paper is sterile at 25C. Another example is the *oz25* allele, which is a non-conditional sterile NECD mutation. Therefore, they did not get a phenotype for those two NECD *lf* allele simply because they used semi-permissive temperatures, weak alleles or conditions (20C) that do not lead to a significant loss of *glp-1* function. If the authors had considered the *gf* alleles, tested more *lf* alleles and/or used more restrictive conditions, they would have likely reached a very different conclusion that disagrees with the statement.
- 2- The preceding statement that "NICD mutations reduce Notch transcriptional activation at the cellular and germline levels while having little impact at the ATS levels" also does not make sense since their data clearly show that all four NICD mutations reduce the Total # of *sygl-1* ATS-containing cells (Fig 2A) and the # *sygl-1* ATS per cell (Fig. 2E). It also needs to be deleted or drastically modified.
- 3- Lines 167-169: I might have missed this on the first pass, but Fig. S4B does show a significant reduction in *let-858* ATS for most *glp-1(lf)* alleles, which is in contradiction with the statement. However, this may be explained by their reduced number of PZ cells. Thus a more informative graph would be the number of ATS per nucleus, which may not be affected and hence better support their statement. Could Fig. 4 data therefore be re-analysed to show the number of ATS per nucleus? Otherwise the statement needs to be toned down (i.e. although there was a reduction in ATS it was not drastic...)
- 4- Lines 181-183, same later for line 271-272: I now understand better the rationale - thank you. However, even if nuclear size gradually increases during G1-S, the average nuclear size would remain the same even with an increased cell cycle rate. Therefore, the statement that "these mutations are unlikely to affect cell cycle progression or its rate" is incorrect. I would suggest to rephrase to "these mutations are unlikely to activate cell cycle checkpoints" as this would indeed lead to an accumulation of cells at a particular cell cycle stage, hence size.
- 5- Lines 280-294: It cannot be said that "changes in the abundance of NICD do not significantly contribute to the regulation of Notch-induced transcriptional activation" since Fig. 6B,C,D,E all clearly show significant ATS or mRNA differences for one or both of the *sel-12* mutations. I agree that the magnitude of the difference is not huge, but this whole paragraph and argument need to be toned down and the caveats need to be pointed out, including the very weak germline phenotype of the *sel-12* alleles used, which suggest they do not impair NICD cleavage very much, and the potential redundancy with *hop-1*. I appreciate that some of the caveats are now pointed out in the discussion, but they are only mentioned after the conclusions are already reached and without apparent consequences. They should be pointed out in the results section and taken into consideration before reaching any conclusion and using it as a section title, etc.

Minor:

- 1- Line 105: Germline "induction" is confusing. Germline "induction" would occur during embryogenesis when the P-lineage is segregated and Z2-Z3 are generated. Replace by "germline function". Or perhaps by "Stemness induction" if this is what was meant?
- 2- Lines 144-145: Inaccurate statements since the *e2142* allele was analysed at 20C in figure 2A but its restrictive temperature is 25C according to Table 1 and reference.

3- Line 151-152: Please add the following (or equivalent) after that sentence. "We however note that we analysed the *glp-1*(e2142) allele at a semi-permissive temperature of 20C and that it is fully sterile at 25C. At that fully restrictive temperature, the *sygl-1* ATS may therefore be lost."

4- Line 161-162: Based on the arguments above, it cannot also be said that "the ANK -and potentially the NICD - plays a more critical role in regulating Notch activity than the NECD". Both the NECD and NICD are important regulators of Notch activity since they are both indispensable for function. Same for Lines 344-346 and the whole paragraph starting with line 371 in the discussion. I appreciate the additions at the end of this paragraph, but the conclusions made (an section title) should be toned down accordingly, which is still not the case.

5- Line 168: a reference is still missing for the statement that *let-858* is a Notch-independent gene constitutively expressed in the germline.

6- Lines 210 and 224: It cannot be said that "each *glp-1* mutation alter the spatial pattern of *sygl-1* transcription distinctively" since no statistical comparisons are made between the mutants in Fig.4. Moreover, and simply by eye, the effects of the e2144 and bn18 alleles look very similar throughout A,B,C.

7- Temperatures still not indicated in Figs.5-6.

Line 281: Based on the arguments above, remove "but not the NECD" from the sentence. Accordingly, the following sentence should be deleted.

Reviewer's Responses to Questions

Experimental quality

Does each figure have the proper controls?

If 'No', please indicate reasons in Comments for Author box below.

Reviewer #1:

- Yes

Reviewer #2:

- Yes

Were the data analyzed using appropriate statistical tests?

If 'No', please indicate reasons in Comments for Author box below.

Reviewer #1:

- Yes

Reviewer #2:

- Yes

Reproducibility

Were experiments performed using adequate number of biological replicates?

If 'No', please indicate reasons in Comments for Author box below.

Reviewer #1:

- Yes

Reviewer #2:

- Yes

Does the methods section provide sufficient detail to permit reproducibility?

If 'No', please indicate reasons in Comments for Author box below.

Reviewer #1:

- Yes

Reviewer #2:

- Yes

Completeness

Are the manuscript's conclusions supported by the data?

If 'No', please indicate reasons in Comments for Author box below.

Reviewer #1:

- No

Reviewer #2:

- No

Scholarship

Do the authors cite and discuss the merits of data that would argue for and against their conclusion?

If 'No', please indicate reasons in Comments for Author box below.

Reviewer #1:

- Yes

Reviewer #2:

- No

Does the manuscript title & abstract accurately reflect the contents of the manuscript, without hyperbole?

If 'No', please indicate reasons in Comments for Author box below.

Reviewer #1:

- Yes

Reviewer #2:

- No

Second revisionAuthor response to reviewers' comments

The revised version of your manuscript was reassessed by the original reviewers. As you will see, the reviewers generally agreed that many of their concerns/issues raised were appropriately addressed; however, some critical points were not. Each reviewer has provided comments on the revisions and have clarified original issues that were not appropriately address. In order to consider this manuscript for publication, several amendments to your manuscript are required. I hope that you will be able to carry these out, because we would like to be able to accept your paper.

Reviewer #1: The authors have satisfactorily addressed all of my major comments. I now only have a few minor points concerning the interpretation of the data in the text that should be revised or discussed. If these changes are incorporated, I do not need to review the manuscript again.

1. The authors replaced some instances of "GSCs" with "Notch-responsive GSCs" as suggested (lines 142-143), but this change was not implemented consistently throughout the manuscript. Please replace "GSCs" with "Notch-responsive GSCs" in all cases where GSCs are identified by using Sygl-1 staining.

We appreciate the reviewer's comment. We have replaced most instances of "GSCs" with "Notch-responsive GSCs." However, we retained a few instances of "GSCs" where the term refers to the entire GSC pool, irrespective of active *sygl-1* transcription.

2. I disagree with the conclusion that NICD mutations play a "more critical role" (line 161) or "have a greater impact on Notch activity" than NECD mutations, which are described as having a "minimal impact." Under restrictive conditions, both NICD and NECD mutants result in infertility. A more accurate conclusion would be that NICD mutations, under partially restrictive conditions, produce a

stronger *lf* effect than NECD. This suggests that the NICD mutations tested are more temperature-sensitive than the NECD mutations. This point could be reframed accordingly, and it may be worth noting as an interesting avenue for future research (line 406).

We appreciate the reviewer's suggestion. We have revised the manuscript (lines 162-163) to moderate the description of NICD's effects relative to NECD.

3. I disagree with the statement that *glp-1(gf)* mutations do not alter Notch transcriptional activation, spatial pattern, or activity at ATS, mRNA levels, and GSC pool size (line 393-394). Figure 5 shows evidence of Notch overactivation, with increased ATS and mRNA extents (Fig. 5A, C, D). Moreover, the manuscript states that the gradient persists in *glp-1(gf)* mutants (lines 410-412), but this is not correct. Specifically, *oz112* does not display a gradient, as shown in Fig. 5B and K-L. We took out the statement about the unaltered spatial pattern (ATS/mRNA extents), which was inconsistent with the data shown in Fig. 5, as the reviewer noted. The lines 410-412 ("Surprisingly, despite the substantial tissue- and cellular-level changes in Notch activation and its activity observed in *glp-1 lf* mutants, the majority of mutations preserved the graded *sygl-1* transcriptional pattern in the distal gonad, although the gradient was not as steep as in wild type (Fig. 4A).") refers specifically to *lf* mutants rather than *gf* mutants; however, we revised it to clarify it.

4. In lines 424-425, the authors claim that the mutations show a strong linear correlation between ATS and mRNA counts per cell. However, in Fig. 7A the R^2 is not strong, and the data distribution could also be consistent with a hyperbolic relationship.

We appreciate the reviewer's observation. We also present individual model fittings for each mutant (Fig. S10), which more accurately depict the linear relationship between ATS and mRNA per cell rather than suggested curved fittings. We have revised the text to reference these individual fittings in Fig. S10 (lines 424-427).

Reviewer #2: This is an improved resubmission. However, it seems like the authors did not give enough importance to one of my comments, so I'll have to be more specific about it. There is indeed one main outstanding issue that I am still uncomfortable with as it would be misleading for someone just reading the abstract. There are a few other outstanding details to fix. Everything can be addressed editorially at this point.

Outstanding issues to address:

1- The statement in the abstract that "NECD mutations have minimal effects across all biological levels" is not true and has to be modified or deleted. Authors themselves agreed that *gf* NECD mutants do have significant effects. In addition, they say that one of the two NECD *lf* mutant that they used in the paper is sterile at 25C. Another example is the *oz25* allele, which is a non-conditional sterile NECD mutation. Therefore, they did not get a phenotype for those two NECD *lf* allele simply because they used semi-permissive temperatures, weak alleles or conditions (20C) that do not lead to a significant loss of *glp-1* function. If the authors had considered the *gf* alleles, tested more *lf* alleles and/or used more restrictive conditions, they would have likely reached a very different conclusion that disagrees with the statement.

We have revised the abstract as requested by the reviewer (lines 20-21 and 26-29) to clarify that our interpretation of the effects of NECD mutations applies only to the partial *lf* mutants analyzed in this study.

2- The preceding statement that "NICD mutations reduce Notch transcriptional activation at the cellular and germline levels while having little impact at the ATS levels" also does not make sense since their data clearly show that all four NICD mutations reduce the Total # of *sygl-1* ATS-containing cells (Fig 2A) and the # *sygl-1* ATS per cell (Fig. 2E). It also needs to be deleted or drastically modified.

We appreciate the reviewer's comment. The term "ATS level" was originally adapted from reviewer 1's suggestion to replace "chromosomal level" in the first review round. To improve clarity, we have now revised the wording to "chromosomal (ATS) level" in line 28. We also revised the manuscript throughout to clearly define and consistently indicate the different biological levels analyzed (germline, cellular, and chromosomal/ATS levels). Both the "total # of *sygl-1* ATS-containing cell (Fig. 2A)" and the "# *sygl-1* ATS per cell (Fig. 2E)" represent Notch readouts at the cellular level rather than at the chromosomal/ATS level, and we also revised the text throughout to make this distinction explicit.

3- Lines 167-169: I might have missed this on the first pass, but Fig. S4B does show a significant reduction in let-858 ATS for most *glp-1(lf)* alleles, which is in contradiction with the statement. However, this may be explained by their reduced number of PZ cells. Thus a more informative graph would be the number of ATS per nucleus, which may not be affected and hence better support their statement. Could Fig. 4 data therefore be re-analysed to show the number of ATS per nucleus? Otherwise the statement needs to be toned down (i.e. although there was a reduction in ATS it was not drastic...)

We appreciate the reviewer's careful reading and interpretation of our data. We have revised the text (lines 168-171) to ensure that it is consistent with the data presented in Fig. S4.

4- Lines 181-183, same later for line 271-272: I now understand better the rationale - thank you. However, even if nuclear size gradually increases during G1-S, the average nuclear size would remain the same even with an increased cell cycle rate. Therefore, the statement that "these mutations are unlikely to affect cell cycle progression or its rate" is incorrect. I would suggest to rephrase to "these mutations are unlikely to activate cell cycle checkpoints" as this would indeed lead to an accumulation of cells at a particular cell cycle stage, hence size.

We appreciate the reviewer's comment. We have revised the texts as suggested.

5- Lines 280-294: It cannot be said that "changes in the abundance of NICD do not significantly contribute to the regulation of Notch-induced transcriptional activation" since Fig. 6B,C,D,E all clearly show significant ATS or mRNA differences for one or both of the *sel-12* mutations. I agree that the magnitude of the difference is not huge, but this whole paragraph and argument need to be toned down and the caveats need to be pointed out, including the very weak germline phenotype of the *sel-12* alleles used, which suggest they do not impair NICD cleavage very much, and the potential redundancy with *hop-1*. I appreciate that some of the caveats are now pointed out in the discussion, but they are only mentioned after the conclusions are already reached and without apparent consequences. They should be pointed out in the results section and taken into consideration before reaching any conclusion and using it as a section title, etc.

We have revised the paragraph (lines 280-296) to tone down our argument as suggested. We also added a sentence stating the redundancy between *sel-12* and *hop-1* (lines 294-296), which is also further discussed in the discussion section.

Minor:

1- Line 105: Germline "induction" is confusing. Germline "induction" would occur during embryogenesis when the P-lineage is segregated and Z2-Z3 are generated. Replace by "germline function". Or perhaps by "Stemness induction" if this is what was meant?

We appreciate the reviewer's comment. We changed it to "germline function."

2- Lines 144-145: Inaccurate statements since the *e2142* allele was analysed at 20C in figure 2A but its restrictive temperature is 25C according to Table 1 and reference.

We thank the reviewer for pointing this out. We have revised the sentence to accurately describe our experimental conditions (lines 144-149).

3- Line 151-152: Please add the following (or equivalent) after that sentence. "We however note that we analysed the *glp-1(e2142)* allele at a semi-permissive temperature of 20C and that it is fully sterile at 25C. At that fully restrictive temperature, the *sygl-1* ATS may therefore be lost."

We have incorporated this statement above into the same paragraph for improve the logical flow (lines 147-149).

4- Line 161-162: Based on the arguments above, it cannot also be said that "the ANK -and potentially the NICD - plays a more critical role in regulating Notch activity than the NECD". Both the NECD and NICD are important regulators of Notch activity since they are both indispensable for function. Same for Lines 344-346 and the whole paragraph starting with line 371 in the discussion. I appreciate the additions at the end of this paragraph, but the conclusions made (an section title) should be toned down accordingly, which is still not the case.

We have revised the sentence as well as throughout the manuscript to tone down our argument as suggested.

5- Line 168: a reference is still missing for the statement that let-858 is a Notch-independent gene constitutively expressed in the germline.

We added a reference.

6- Lines 210 and 224: It cannot be said that "each *glp-1* mutation alter the spatial pattern of *sygl-1* transcription distinctively" since no statistical comparisons are made between the mutants in Fig.4. Moreover, and simply by eye, the effects of the *e2144* and *bn18* alleles look very similar throughout A,B,C.

We revised the title to "*glp-1* mutations alter the spatial pattern of *sygl-1* transcriptional activation."

7- Temperatures still not indicated in Figs.5-6.

Line 281: Based on the arguments above, remove "but not the NECD" from the sentence. Accordingly, the following sentence should be deleted.

Temperatures (all 20 °C) are indicated in the figure legends. The sentence has been revised as suggested above by the reviewer.

Third decision letter

MS ID#: bio.062008R2

MS Title: Genetic mutations in GLP-1/Notch pathway reveal distinct mechanisms of Notch signaling in germline stem cell regulation

Authors: ChangHwan Lee, Nimmy Sara John, Michelle Alexandra Urman, Mahasin Gulnaaz Mehmood and Vanessa Gentile

I am happy to tell you that your manuscript has been accepted for publication in Biology Open, pending our standard publication integrity checks. It was accepted on 11th December 2025.